# Sensory ataxia and cardiac hypertrophy caused by neurovascular oxidative stress in chemogenetic transgenic mouse lines

Shambhu Yadav [1], Markus Waldeck-Weiermair[1], Fotios Spyropoulos [1,2], Roderick Bronson[3], Arvind K. Pandey [1], Apabrita Ayan Das[1], Alexander C. Sisti[1], Taylor A. Covington[1], Venkata Thulabandu [1], Shari Caplan [4], William Chutkow[4], Benjamin Steinhorn[1] & Thomas Michel [1] ✉

Oxidative stress is associated with cardiovascular and neurodegenerative diseases. Here we report studies of neurovascular oxidative stress in chemogenetic transgenic mouse lines expressing yeast D-amino acid oxidase (DAAO) in neurons and vascular endothelium. When these transgenic mice are fed D-amino acids, DAAO generates hydrogen peroxide in target tissues. DAAO-TG[Cdh5] transgenic mice express DAAO under control of the putatively endothelial-specific Cdh5 promoter. When we provide these mice with D-alanine, they rapidly develop sensory ataxia caused by oxidative stress and mitochondrial dysfunction in neurons within dorsal root ganglia and nodose ganglia innervating the heart. DAAO-TG[Cdh5] mice also develop cardiac hypertrophy after chronic chemogenetic oxidative stress. This combination of ataxia, mitochondrial dysfunction, and cardiac hypertrophy is similar to findings in patients with Friedreich's ataxia. Our observations indicate that neurovascular oxidative stress is sufficient to cause sensory ataxia and cardiac hypertrophy. Studies of DAAO-TG[Cdh5] mice could provide mechanistic insights into Friedreich's ataxia.

Oxidative stress is a hallmark of neurodegeneration and has long been associated with elevated levels of reactive oxygen species (ROS) both in the vasculature and the central nervous system[1–6]. Our "chemogenetic" approach enables us to test the hypothesis that oxidative stress plays a causal role in disease pathogenesis by dynamically and specifically modulating ROS in target tissues in vivo using the yeast enzyme D-amino acid oxidase (DAAO) to generate the stable reactive oxygen species hydrogen peroxide ($H_2O_2$)[7]. DAAO synthesizes $H_2O_2$ when provided with its D-amino acid substrate, but this yeast enzyme is quiescent when expressed in untreated mammalian cells—which typically contain only L-amino acids. When cells or animals expressing yeast DAAO are provided with D-alanine, the intracellular $H_2O_2$ generated by DAAO causes oxidative stress[8]. We previously used this chemogenetic approach to create a model of heart failure caused by chemogenetic oxidative stress[7]: animals expressing DAAO in cardiac myocytes develop heart failure when D-alanine is provided in their drinking water[9]. In the present studies, we extend this in vivo chemogenetic approach to generate oxidative stress in the vascular endothelium and in neurons (hence the term "neurovascular" used here) exploiting transgenic/chemogenetic mouse lines that express DAAO in these tissues. Our characterizations of these transgenic/chemogenetic mouse models reveal unexpected phenotypes of neurodegeneration, mitochondrial dysfunction, and cardiac hypertrophy caused by neurovascular oxidative stress.

[1]Division of Cardiovascular Medicine, Department of Medicine, Brigham and Women's Hospital, Harvard Medical School, 75 Francis Street, Boston, MA 02115, USA. [2]Department of Pediatric Newborn Medicine, Brigham and Women's Hospital, Harvard Medical School, 75 Francis Street, Boston, MA 02115, USA. [3]Department of Immunology, Harvard Medical School, Boston, MA 02115, USA. [4]Novartis Institutes for Biomedical Research, Cambridge, MA 02139, USA. ✉e-mail: thomas_michel@hms.harvard.edu

## Results

### Chemogenetic DAAO-TG$^{Cdh5}$ mice

We designed and constructed a chemogenetic/transgenic mouse line with the primary goal of expressing DAAO in vascular endothelial cells to study the effects of in vivo oxidative stress on vascular disease states. The transgenic construct that we developed (shown in Supplementary Fig. 1a) consists of a fusion protein between DAAO and the fluorescent ratiometric $H_2O_2$ biosensor HyPer[10]. This enables us both to generate $H_2O_2$ (by providing D-amino acids to DAAO) and to detect $H_2O_2$ (by ratiometric imaging of HyPer) in the transgenic animals or in isolated cells or tissues. This transgenic construct contains a stop codon flanked by LoxP sites and was targeted to the Rosa26 locus using CRISPR/Cas9 methods. After verifying appropriate transgene insertion at the Rosa26 locus and generating founder lines, we crossed this conditionally inducible DAAO transgenic line with an established mouse line expressing Cre recombinase under the control of the VE-cadherin (Cdh5) promoter[11] to create DAAO-TG$^{Cdh5}$ mouse lines. The Cdh5 promoter has been extensively characterized as an endothelial cell-specific promoter, and has been used to construct numerous transgenic mouse lines to study endothelium-specific gene expression[11]. We isolated vascular tissues from transgene-positive DAAO-TG$^{Cdh5}$ animals and performed immunoblots probed with antibodies against GFP, which detects the HyPer within the HyPer-DAAO fusion protein. We confirmed recombinant protein expression in vascular tissues isolated from DAAO-TG$^{Cdh5}$ transgene-positive animals; littermates expressing only the Cre recombinase (Cre$^+$/TG$^-$) showed no transgene expression (Supplementary Fig. 1b).

### Sensory ataxia in DAAO-TG$^{Cdh5}$ mice

We had expected that the DAAO-TG$^{Cdh5}$ mouse line would yield a phenotype in the vasculature in response to chronic D-alanine feeding. But when we added D-alanine to the animals' drinking water to activate the transgenic DAAO enzyme in endothelial cells, we found to our surprise that the DAAO-TG$^{Cdh5}$ mice were unable to walk within a few days. The behavior of control mice (littermates confirmed to be Cre-positive and transgene-negative, termed "Cre$^+$/TG$^-$") were unaffected by D-alanine feeding. The systolic blood pressure (measured using tail cuff) of the D-alanine-fed DAAO-TG$^{Cdh5}$ transgenic mice (95 ± 5 mm) was not different from the blood pressure in D-alanine fed Cre$^+$/TG$^-$ control mice (100 ± 10 mm; $n = 3$ each group, $p = 0.5$). DAAO-TG$^{Cdh5}$ mice developed a profound ataxia that started in the hindlimbs and progressed over several days to the forelimbs (Fig. 1, Supplementary Movie 1 and Supplementary Fig. 2a). It was not clear whether the striking locomotory defects that we observed in the D-alanine-fed transgenic animals derived from problems arising in the animals' blood vessels, muscles, nervous system, or elsewhere. Given that the Cdh5-driven Cre-line has been well-characterized for its endothelial expression, we initially hypothesized that the ataxia phenotype was a consequence of pathological oxidative stress on the vasculature. Indeed, we observed robust expression of the transgene protein in the vascular endothelium of vascular tissues isolated from untreated DAAO-TG$^{Cdh5}$ mice (Fig. 2a). However, when we performed necropsies on the affected animals, we found that the blood vessels were apparently normal, with no obvious signs of vascular pathology. The most striking abnormality detected on necropsy was that the dorsal column of the spinal cord, which transmits sensory signals, had degenerated (Fig. 1a–c). In contrast, the ventral structures were apparently unaffected (Supplementary Fig. 1c, d); peripheral nerves in the hindlimb skeletal muscle also showed signs of degeneration, while the skeletal muscle itself was normal. We then performed a broad range of behavioral tests[12] to further characterize the ataxia observed in these mice. For these studies, we compared the behavior of D-alanine-fed DAAO-TG$^{Cdh5}$ mice with their transgene-negative D-alanine-fed littermates expressing Cre recombinase (Cre$^+$/TG$^-$,

identified by PCR) serving as controls. D-alanine feeding of the DAAO-TG$^{Cdh5}$ mice led to a rapid and profound decrease in their locomotor activity, coordination, and performance across a wide range of behavioral tests (Fig. 1d–j and Supplementary Fig. 2a), which persisted after cessation of D-alanine feeding (Supplementary Fig. 2b). When DAAO-TG$^{Cdh5}$ mice were treated first with an antioxidant "cocktail" consisting of N-acetyl cysteine (10 mM) and sodium selenite (10 μM)[13–15], the ataxia phenotype induced by D-alanine was attenuated.

### Neurodegeneration in dorsal root ganglia

Correct insertion of the DAAO-TG$^{LoxP}$ transgene into the Rosa26 locus[16] had been confirmed in the founder transgenic lines, and there was robust transgene expression in the vascular endothelium in the DAAO-TG$^{Cdh5}$ offspring (Fig. 2a). We were determined to discover how such a striking and highly specific neuropathological phenotype would appear in an "endothelial cell-specific" transgenic mouse. We first sought to identify the neuronal cell(s) that might be responsible for the sensory tract degeneration that we had observed in the spinal cord of D-alanine-treated DAAO-TG$^{Cdh5}$ mice. We isolated dorsal root ganglia (DRG) from the lumbar region of the spinal cord (Fig. 2b). The DRG contain the neuronal cell bodies that send axons to the lumbar dorsal spinal cord tracts, which had degenerated upon feeding D-alanine to the DAAO-TG$^{Cdh5}$ mice (Fig. 1b, c). We observed robust transgene expression in nearly all of the DRG neurons (Fig. 2c). Silver staining of the DRG revealed dead and dying neurons (Fig. 2d), and electron microscopy of isolated DRG neurons showed markedly distorted mitochondria with bizarrely disrupted mitochondrial cristae (Fig. 2e) following D-alanine feeding of the DAAO-TG$^{Cdh5}$ mice, while the mitochondria from DRG of D-alanine-fed control mice had normal morphology (Fig. 2f). We analyzed the mitochondrial membrane potential in DRG isolated from D-alanine-fed mice (Supplementary Data Fig. 2c), which revealed that mitochondrial DRG from D-alanine-treated DAAO-TG$^{Cdh5}$ mice had membrane potential that was significantly lower than the DRG mitochondria of D-alanine-fed control mice, providing evidence of mitochondrial dysfunction in the DRG from D-alanine-treated DAAO-TG$^{Cdh5}$.

To explore the effects of chronic chemogenetic oxidative stress on gene expression in DAAO-TG$^{Cdh5}$ mice, we performed RNA sequencing on DRG isolated from DAAO-TG$^{Cdh5}$ mice following longer-term (6 weeks) D-alanine feeding. Again, Cre$^+$/TG$^-$ littermates served as controls. Due to the rapid onset of incapacitating ataxia that resulted from 0.75 M D-alanine feeding—which precludes longer-term studies—we instead used a lower concentration of D-alanine in the animals' drinking water (0.5 M). Transgenic DAAO-TG$^{Cdh5}$ mice fed 0.5 M D-alanine still developed ataxia, but over a period of weeks instead of days; the control Cre$^+$/TG$^-$ littermates did not develop ataxia. The ataxia phenotype and the pattern of DRG degeneration seen after 6 weeks of lower-dose D-alanine feeding was identical to what had been seen after 4–6 days of higher-dose D-alanine treatment (Figs. 1 and 2). After 6 weeks of D-alanine (0.5 M) feeding, animals were sacrificed, and lumbar DRG were isolated from each mouse and processed for RNA sequencing[17]. RNA sequence analysis (Fig. 3 and Supplementary Figs. 3 and 4) revealed that 144 transcripts (out of 13,380 total transcripts) were significantly more abundant in DRG from the D-alanine-fed DAAO-TG$^{Cdh5}$ mice than in DRG from control mice fed D-alanine; 39 transcripts were less abundant in DRG from the transgenic animals, and the vast majority of transcripts (13,197) did not show a significant change in abundance. Gene Ontology analyses revealed that most of the differentially expressed upregulated transcripts are implicated in immune response and/or inflammatory pathways (Supplementary Fig. 4). We imputed the cells of origin of these transcripts based on the recently published single-cell RNA sequencing data in mouse DRG[18], which allowed us to assign the probable cells of origin based by overlaying our bulk RNA-seq data over the single-cell dataset, as

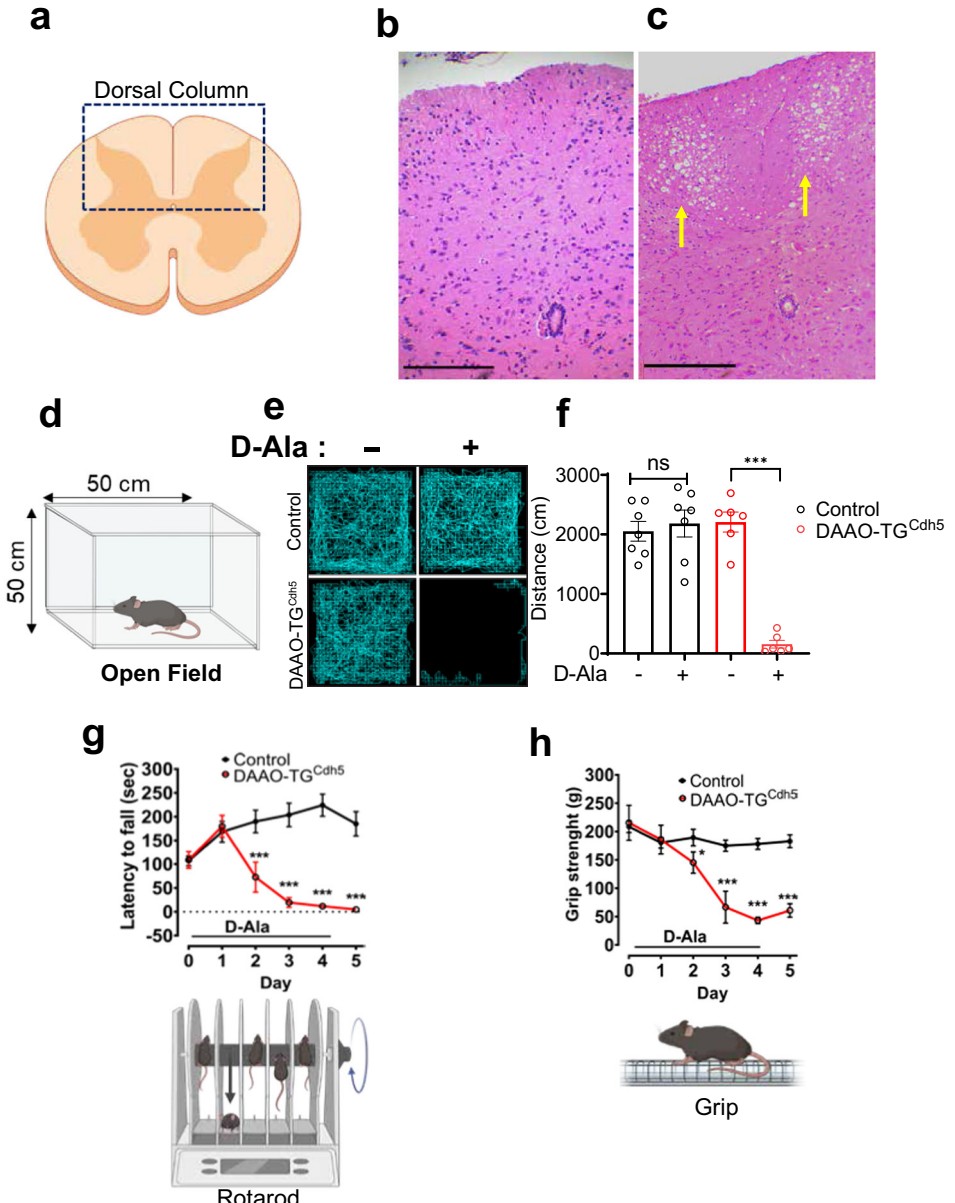

**Fig. 1 | Neuropathological and neurobehavioral features of D-alanine-fed DAAO-TG<sup>Cdh5</sup> mice. a** shows a pictorial representation of a transverse section through the mouse spinal cord noting the dorsal column. **b, c** show representative haematoxylin/eosin staining of a transverse section of the lumbar spinal cord isolated from control (**b**), $n = 3$ biologically independent Cre⁺/TG⁻ mice and DAAO-TG<sup>Cdh5</sup> (**c**) mice that had been treated for 6 days with D-alanine, revealing marked and highly selective degeneration of the dorsal column (noted by the arrows) only in the transgenic mouse, $n = 3$ biologically independent mice. Scale bars indicate 100 µm. **d** shows a pictorial representation of the open field mouse behavioral test[60], in which mice are allowed to move freely for a fixed amount of time while their activity is quantitated by infrared beams mounted in the apparatus. **e** shows quantitation of field activity traces for a representative DAAO-TG<sup>Cdh5</sup> mouse and a Cre⁺/TG⁻ control mouse before and after providing D-alanine (D-Ala) in their drinking water for 4 days, as noted; **f** presents the results of summary data for this behavioral assay obtained for $n = 7$ DAAO-TG<sup>Cdh5</sup> mice and $n = 6$ control mice. Data are presented as mean values ± SEM; ***denotes $p < 0.001$ (by two-way ANOVA); **g** presents a pictorial representation of the Rotarod test[60], in which mice are placed on a slowly rotating rod and the time until they fall off the rod (latency to fall) is monitored as an index of balance and coordination. Both DAAO-TG<sup>Cdh5</sup> mice and littermate control Cre⁺/TG⁻ mice were treated with D-alanine in the drinking water and tested daily; a pictorial representation of the apparatus is shown below, and summary results are shown above. Data are presented as mean values ± SEM; ***denotes $p < 0.001$ (by two-way ANOVA); $n = 7$ DAAO-TG<sup>Cdh5</sup> and $n = 6$ Cre⁺/TG⁻ control mice each group. **h** shows the results of grip strength testing[61] of DAAO-TG<sup>Cdh5</sup> and littermate Cre⁺/TG⁻ control mice treated with D-alanine; a pictorial representation of the testing apparatus is shown below. Data are presented as mean values ± SEM; ***denotes $p < 0.001$ (two-way ANOVA) and $n = 7$ DAAO-TG<sup>Cdh5</sup> and $n = 6$ Cre⁺/TG⁻ mice were studied in each group.

we have done previously in the heart[9]. We found that most of the upregulated transcripts are specific for macrophages or satellite glial cells (Fig. 3e). By contrast, most of the downregulated transcripts (Supplementary Fig. 3a) can be assigned to DRG fibroblasts, Schwann cells, or endothelial cells (Supplementary Fig. 3b). GSEA[19] biological process enrichment showed genes related to immune response pathways; enrichment of cellular component analysis mainly reflect

extracellular matrix genes where molecular function enrichment showed increased abundance of immune response genes (Supplementary Fig. 4a–c).

We performed proteomic analyses of DRG isolated from D-alanine-treated DAAO-TG<sup>Cdh5</sup> mice and their control littermates (Fig. 4 and Supplementary Fig. 5). As we found in our transcriptomic profiling, these proteomic analyses revealed changes in multiple

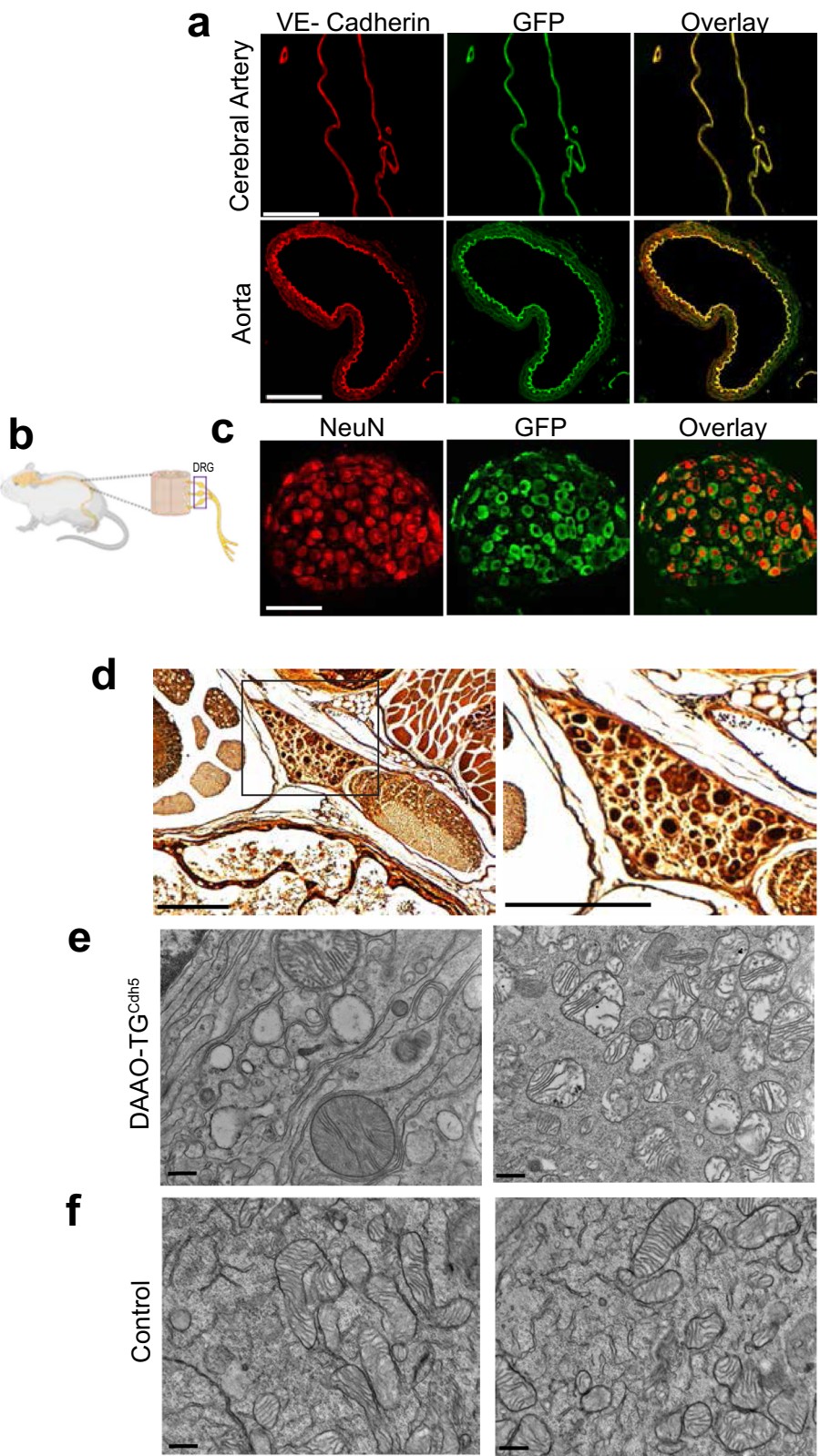

inflammation-related pathways (Supplementary Fig. 5). Behavioral phenotype analysis (MGI)[20,21] identified the most significant phenotypic changes related to only three principal classes: movement disorder, nerve degeneration, and muscle pathology (Fig. 4). KEGG metabolic pathway analyses[22,23] showed enrichment of neurodegeneration-, metabolic-, and oxidative stress-related pathways in DRG from D-alanine-fed DAAO-TG^Cdh5 mice, as well as changes in diverse signaling and inflammation-related pathways. The proteomics signature thus recapitulates nearly all key elements of the neuropathic phenotype (Figs. 1 and 2), and also raises an intriguing connection to muscle pathology that we cautiously anticipated given the many parallels between these mice and the symptoms found in patients with the inherited disease, Friedreich's ataxia (FA). These connections are explored in the experiments shown below.

**Fig. 2 | DAAO-TG$^{Cdh5}$ transgene expression in blood vessels and DRG neurons.**
**a** shows representative immunofluorescence images of arterial preparations (from aorta or middle cerebral artery, as noted) that were isolated from untreated DAAO-TG$^{Cdh5}$ mice, and then fixed and stained as indicated with antibodies directed against VE-cadherin (endothelial cell marker) or GFP (which detects the YFP contained in the HyPer-DAAO fusion protein transgene). The scale bars indicate 100 μm. **b** shows a pictorial representation of the dorsal root ganglia (DRG) located adjacent to the lumbar spinal cord in the mouse. **c** shows representative immunofluorescence images of DRG isolated from untreated DAAO-TG$^{Cdh5}$ mice and stained with antibodies directed against the neuronal marker NeuN[67] or with GFP

antibodies to detect the transgene, as indicated. The scale bars note 100 μm.
**d** shows a photomicrograph of silver-stained fixed DRG that were isolated from D-alanine-treated DAAO-TG$^{Cdh5}$ mice; the dark staining is indicative of neuronal death[68] (The right-side figure is at a higher magnification of the inset in the left-side figure). The scale bars indicate 100 μm. **e** shows electron microscopic images of lumbar DRG isolated from D-alanine-treated TG$^{Cdh5}$ mice, revealing swollen and distorted mitochondria. The scale bars indicate 500 nm. **f** shows electron microscope images of lumbar DRG isolated from D-alanine-treated control mice. The scale bars indicate 500 nm. All images are representative of $n = 3$ animals per group.

## Oxidative stress and cardiac hypertrophy

The constellation of sensory ataxia, DRG degeneration, oxidative stress, and mitochondrial disruption is quite similar to the phenotypic features characteristic of patients with Friedreich's ataxia[15]. FA is the most common inherited ataxia in patients, and is an autosomal recessive disease caused by mutations in the frataxin gene[24]. Patients with FA develop a progressive sensory ataxia in childhood, but typically die from heart failure and malignant arrhythmias as a consequence of progressive cardiac hypertrophy[25,26]. The mechanisms whereby FA causes cardiac hypertrophy are incompletely understood, but it occurs later in the time course of disease in most patients, and to provide sufficient time for cardiac remodeling we therefore decided to treat the animals for a longer time with a lower dose of D-alanine than had been used in the short-term experiments (Figs. 1 and 2). To determine if there is a cardiac phenotype in the DAAO-TG$^{Cdh5}$ mice, we performed echocardiography on transgenic and control mice chronically treated with D-alanine (0.5 M for 6 weeks). Quantitative analyses of echocardiographic parameters (performed by blinded sonographers) in DAAO-TG$^{Cdh5}$ mice fed D-alanine revealed a significant increase in left ventricular ejection fraction and fractional shortening for DAAO-TG$^{Cdh5}$ mice relative to control mice, accompanied by a decrease in left ventricular systolic and diastolic volumes and a significant increase in posterior ventricular wall thickness (Fig. 5a, b). These findings are consistent with development of cardiac hypertrophy in DAAO-TG$^{Cdh5}$ mice following chronic D-alanine feeding compared to control animals. In order to determine whether the transgene is expressed in cardiac myocytes, we stained heart sections with the GFP antibody (which detects the YFP sequence contained in the HyPer-DAAO transgene fusion protein), and found no signal for the transgene in cardiac myocytes, while nearby coronary blood vessels had robust transgene expression in the vascular endothelium (Fig. 5c).

Based on our observations of neuronal expression of the DAAO transgene, we hypothesized that perturbations in cardiac innervation may play a central role in the cardiomyopathy seen in D-alanine fed DAAO-TG$^{Cdh5}$ mice. We extended our analyses to include the major peripheral nerve ganglia that control cardiac function (Fig. 5d): the stellate ganglion (containing sympathetic efferent neurons) and nodose ganglion (containing parasympathetic afferent neurons)[27,28]. As shown in Fig. 5e, silver staining of nodose ganglia isolated from D-alanine-fed DAAO-TG$^{Cdh5}$ mice reveals signs of striking neurodegeneration, but nodose ganglion from the D-alanine-fed Cre+/TG$^-$ control mice show no neurodegeneration. As can be seen in Fig. 5f, nearly all the neurons in the nodose ganglion express the transgene, while a minority of nodose ganglion neurons stain positive for tyrosine hydroxylase, as previously reported[29]. By contrast, the stellate ganglion shows no transgene expression in neurons (Fig. 5g). Most of the neurons in the stellate ganglion stain positive for tyrosine hydroxylase (Fig. 5g), as has been reported[29]. Thus, the two major peripheral nerve ganglia controlling the heart differ markedly in their pattern of transgene expression and neurodegeneration: the nodose ganglion neurons show robust transgene expression and undergo degeneration after D-alanine feeding, while the stellate ganglion neurons demonstrate neither transgene expression nor neurodegeneration.

The unexpected neurological and cardiac phenotypes of DAAO-TG$^{Cdh5}$ mice led us to generate and characterize a second "endothelium-specific" mouse line in which transgene expression would be driven by a distinct endothelial promoter. To dissect out primary endothelial responses from primary neuronal responses, we employed the extensively-characterized putatively endothelial cell-specific Tie2 promoter[30]. We crossed the DAAO-TG$^{LoxP}$ line with a commercially available Tie2-Cre line (Jackson Labs) and identified founder lines, which were then characterized. Vascular preparations from DAAO-TG$^{Tie2}$ mice showed robust functional transgene expression in mesenteric and coronary arteries (Supplementary Fig. 6a, c), but there was no transgene expression in DRG neurons (Supplementary Fig. 7). Treatment of DAAO-TG$^{Tie2}$ mice either with high dose (0.75 M) or low dose (0.5 M) D-alanine feeding did not result in ataxia, and there also was no cardiac phenotype assessed by echocardiography. We generated a DAAO-TG$^{Car}$ transgenic mouse line expressing the HyPer-DAAO fusion protein in cardiac myocytes under control of the cardiac-specific Myh6 (myosin heavy chain) promoter[31,32]. We performed HyPer ratio imaging for $H_2O_2$, and found robust functional transgene expression in DAAO-TG$^{Car}$ cardiac myocytes, with no evidence of transgene expression in cardiac myocytes isolated from the DAAO-TG$^{Cdh5}$ line (Supplementary Fig. 6d). In contrast, both the DAAO-TG$^{Car}$ and DAAO-TG$^{Tie2}$ mouse lines had robust $H_2O_2$ responses to D-alanine in isolated coronary arteries (Supplementary Fig. 6e).

## Discussion

Oxidative stress has long been associated with a wide range of disease states ranging from atherosclerosis to heart failure to neurodegeneration[33,34]. What has been less clear is whether oxidative stress is causal, or is merely associated with these disease phenotypes. Chemogenetic approaches using recombinant DAAO have proven highly informative in establishing causality[8]. By dynamically controlling the generation of reactive oxygen species through providing D-alanine to DAAO (with appropriate controls), the present studies provide evidence establishing that oxidative stress from $H_2O_2$ is sufficient to cause selective degeneration of sensory neurons, leading to a striking sensory ataxia (Fig. 1). To be sure, oxidative stress may play only a more limited role in the pathobiology of other neurodegenerative disease states, but the dependence of DRG degeneration on both the DAAO transgene and D-alanine feeding provides strong evidence that oxidative stress is causal in generating this neuropathic phenotype. The level of $H_2O_2$ generated by DAAO is within the range of $H_2O_2$ concentrations seen within normal cells and in disease states[6,35–37]. The attenuation of the ataxia phenotype by the co-administration of oral antioxidants provides further support for this hypothesis (Supplementary Fig. 2).

We had not expected a neuropathic phenotype when we first generated transgenic mice expressing DAAO under control of the putatively endothelial cell-specific Cdh5 promoter, which has been widely used to generate mouse lines expressing transgenes in endothelial cells[11]. A recent map of DRG promoter activity in embryonic development has reported that the Cdh5 promoter is transiently active early in fetal development, but the Cdh5 promoter does not remain active in the DRG of adult mice[38], as we found in the

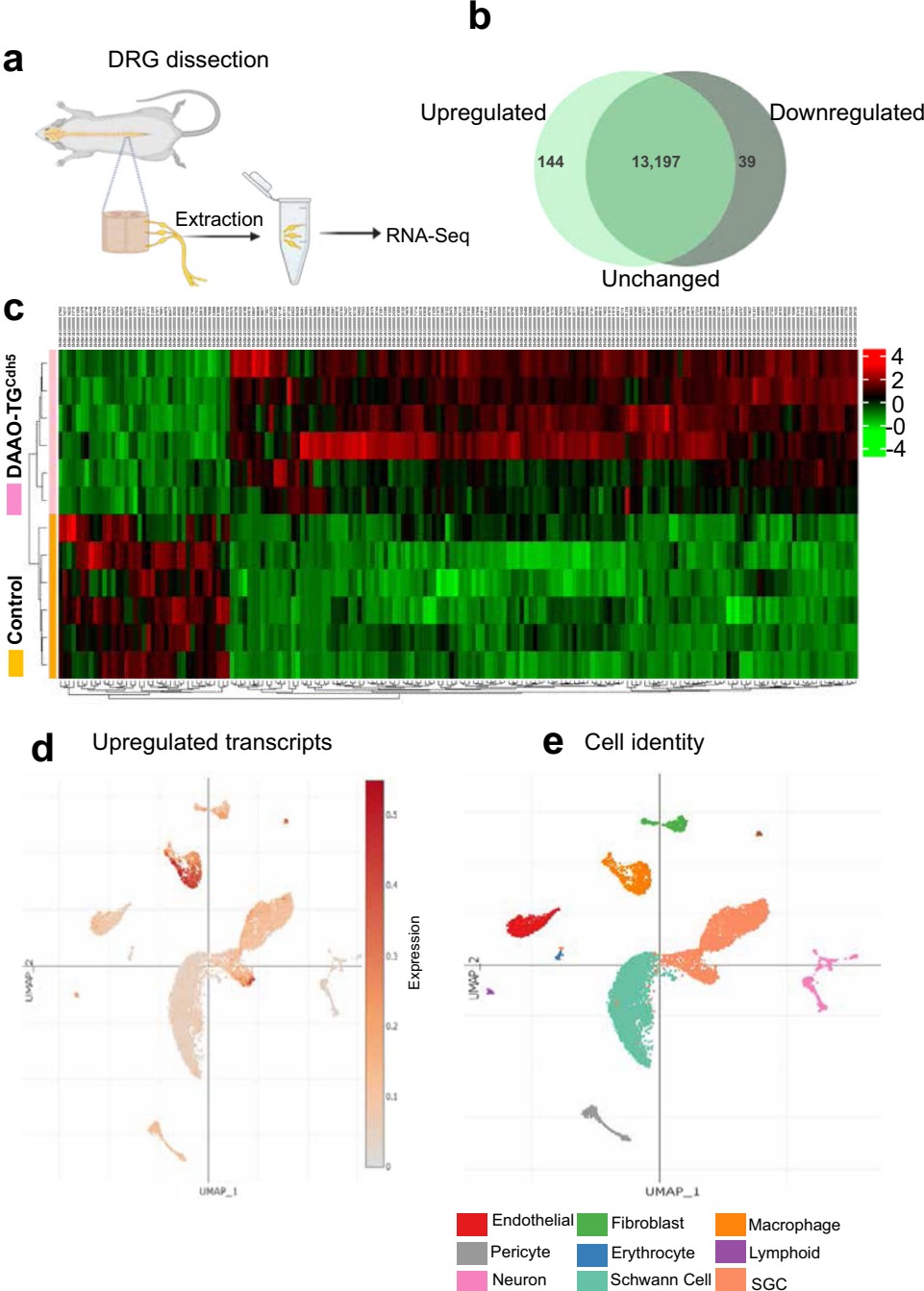

**Fig. 3 | Transcriptional profiling of DRG from D-alanine-treated DAAO-TG^Cdh5.**
**a** shows a schematic of the experimental approach to RNA sequencing of DRG isolated from DAAO-TG^Cdh5 mice or control Cre+/TG− littermates treated with D-alanine (0.5 M) for 7 weeks. **b** shows a Venn diagram presenting the numbers of upregulated, downregulated, and unchanged individual transcripts present in DRG isolated from D-alanine-treated DAAO-TG^Cdh5 mice relative to the abundance of these transcripts in control littermates, **c** shows a heat map displaying significantly up- and downregulated transcript levels among the individual DAAO-TG^Cdh5 or control Cre+/TG− mice treated with D-alanine (six independent biological replicates for each genotype). The red color indicates an increase in abundance for individual transcripts, and the green color represents decreased transcript levels[63] comparing DAAO-TG^Cdh5 mice with control Cre+/TG− mice, as indicated. **d** shows a uniform manifold approximation and projection (UMAP) plot representing the upregulated (≥2 fold) transcripts in DRG superimposed over the global uniform manifold approximation and projection distribution; significantly increased transcripts are noted in red. **e** shows the global UMAP plot of the different cell populations of the DRG. Each dot represents an individual cell, and the colors represent the cells' respective subcluster, as noted at the bottom of the figure. The cellular origin of the upregulated DRG transcripts can be imputed by overlaying the bulk RNA sequencing data obtained in the present studies (**d**) with recent single-cell RNA sequencing data from Jager et al.[18] reported in the single-cell portal (Single-Cell Portal: https://singlecell.broadinstitute.org/single_cell) (**e**).

present studies. Nonetheless, Cdh5-driven Cre expression in fetal DRG appears to be sufficient to excise the stop codon from the DAAO-TG^LoxP and thereby permit expression of the transgene in DRG as well as in vascular endothelium in adult DAAO-TG^Cdh5 mice, leading to the striking phenotype seen in these studies. Most if not all of the

neurons in the DRG of mice show transgene expression, and it is possible that other sensory perceptions (e.g., pain, touch) are also affected, but the severe ataxia in these mice undermines more detailed behavioral analyses. Expression of Cdh5 in sensory neurons may raise caveats about the interpretation of the many previous

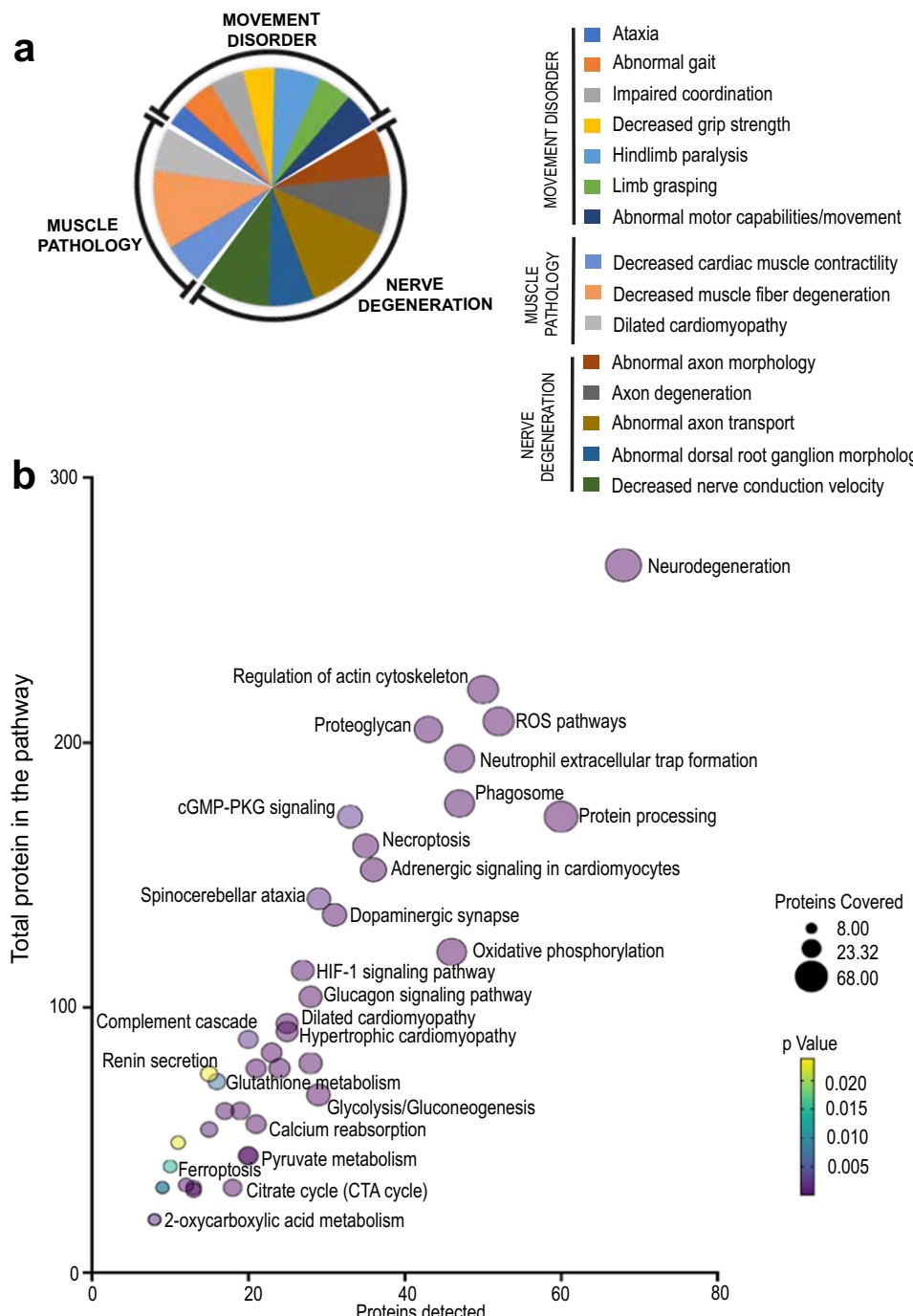

**Fig. 4 | Proteomic analyses of isolated DRG from D-alanine-fed DAAO-TG<sup>Cdh5</sup> and Cre<sup>+</sup>/TG<sup>−</sup> control mice. a** shows the results of phenotypic enrichment analysis using GeneCodis 4.0 to assign proteomic patterns to mouse phenotypes[20,21]. The legend on the right lists the principal phenotypes detected by GeneCodis, which we have then assigned post hoc to three major phenotype classes represented by these abnormalities: movement disorder; nerve degeneration; and muscle pathology. Each color represents a separate subgroup under each class, with the size of each slice corresponding to the corresponding percent enrichment for that phenotype, calculated with respect to the total enrichment score. **b** shows pathway enrichment analysis with KEGG by GeneCodis 4.0[22,23] shown in a bubble plot indicating

significant enrichment of neurodegeneration, metabolic, and oxidative stress-related pathways, as well as diverse signaling and inflammation-related pathways. In this plot, the number of genes covered in each pathway is represented by the size in the bubble position as well as its position on the abscissa; the total number of genes present in the pathway is shown on the ordinate; the color of the bubble corresponds to the *p* value, as noted. For the KEGG pathway analyses hypergeometric test was used to calculate the enrichment and *p* value of each pathway. Adjustment for the multiple comparison was performed to obtain adjusted *p* values for each pathway enriched from the data.

reports that relied upon the endothelial specificity of Cdh5-driven transgenic mouse lines. However, this concern does not apply to studies of transgenic mice that used inducible Cdh5 promoters that are activated only in adult mice, which do not exhibit Cdh5 promoter activity in neurons[30,38].

The striking ataxia seen in the DAAO-TG<sup>Cdh5</sup> mice in these studies undermines longer-term studies of this mouse line: the animals become incapacitated within days to a few weeks of starting D-alanine feeding and need to be euthanized, precluding studies of the chronic effects of vascular oxidative stress. We therefore focused our studies

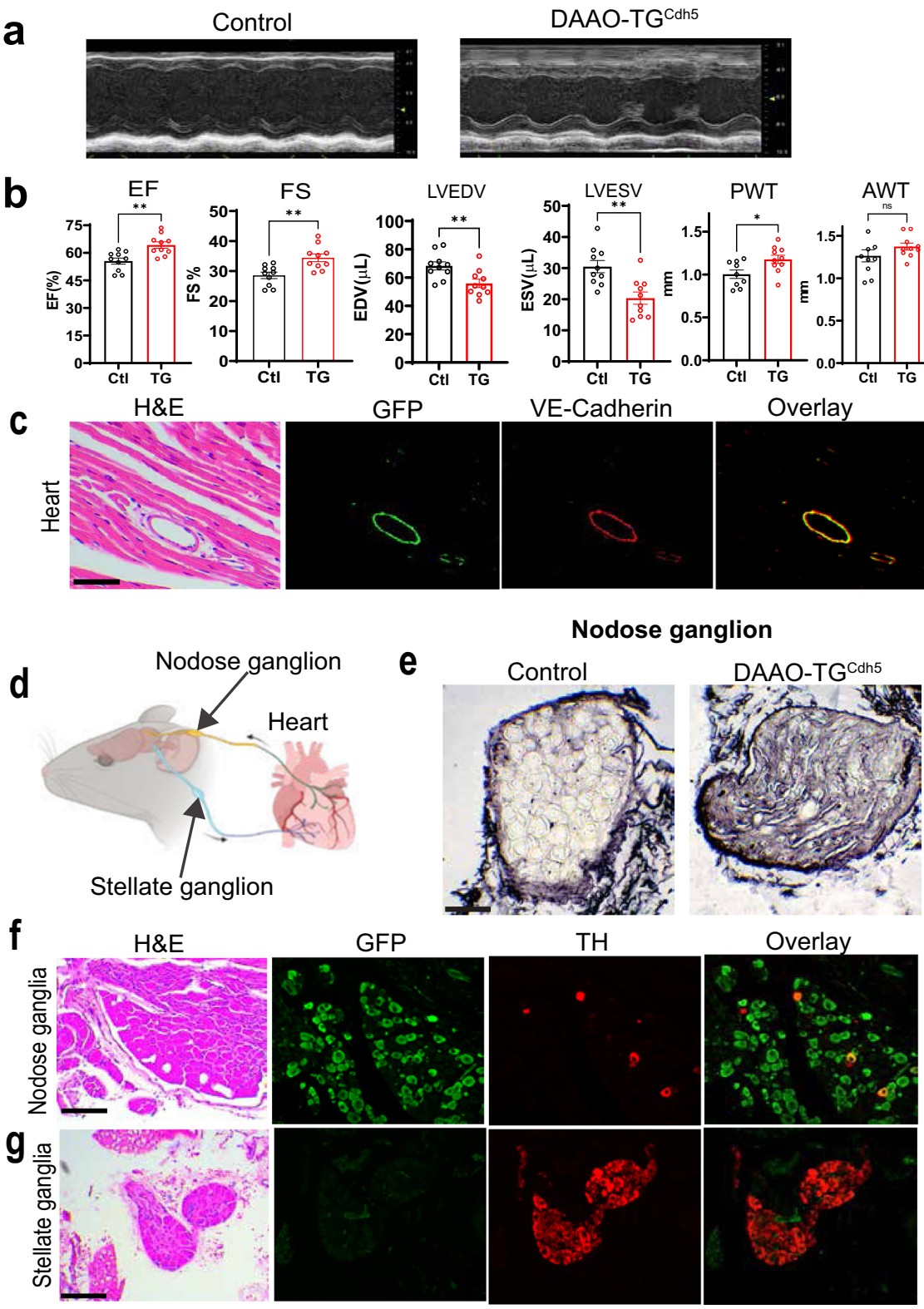

on identifying the molecular and cellular pathways that are acutely involved in DRG neurodegeneration as a consequence of oxidative stress. RNA sequencing in DRG isolated from D-alanine-treated DAAO-TG[Cdh5] mice revealed a striking increase in transcripts that are characteristic of inflammation-related responses (Fig. 3 and Supplementary Fig. 3). We used published single-cell RNA sequencing data from DRG[18] to impute that these transcriptional changes reflect an increase in the abundance of macrophages and satellite glial cells in DRG isolated

from DAAO-TG[Cdh5] transgenic mice following oxidative stress. These data suggest that chemogenetic oxidative stress in DRG leads to an infiltration of inflammatory response cells, associated with neuronal degeneration and cell death. The results of proteomic analyses (Fig. 4 and Supplementary Fig. 5) complement and extend these transcriptomic findings. Phenotypic prediction analysis from annotated DRG proteins reveals that the most important predicted phenotypes relate to movement disorder, nerve degeneration, and muscle

**Fig. 5 | Cardiac hypertrophy and ganglion-specific neuronal degeneration after induction of chemogenetic oxidative stress in DAAO-TG^Cdh5 mice. a** presents representative echocardiographic images showing short-axis M-mode views of the left ventricle of D-alanine-fed (0.5 M D-alanine for 6 weeks) DAAO-TG^Cdh5 mice and Cre⁺/TG⁻ control mice. **b** shows quantitative analyses of echocardiographic parameters from these mice that were quantitated by observers blinded to genotype and treatment. Measurements include left ventricular ejection fraction (EF; $p = 0.0039$), fractional shortening (FS; $p = 0.0039$), left ventricular end-diastolic volume (LVEDV; $p = 0.0089$), left ventricular end-systolic volume (LVESV; $p = 0.0021$), posterior wall thickness (PWT; $p = 0.0279$), and anterior wall thickness (AWT; $p = 0.1655$). *denotes $p < 0.05$; **denotes $p < 0.01$; and ***denotes $p < 0.001$ (Mann–Whitney test, unpaired two-tailed). Data are presented as mean ± standard error; there were $n = 10$ animals per group (for posterior wall thickness $n = 9$ per group). **c** shows representative images from fixed mouse heart tissue isolated from untreated DAAO-TG^Cdh5 mice; the first panel shows H&E staining; the next panels show (as noted) immunofluorescent staining with GFP antibodies (to detect the transgene) and VE-Cadherin antibodies (endothelial marker) followed by an overlay of the GFP and VE-Cadherin signals. The scale bars note 100 μm. Expression of the transgene is clearly seen in the blood vessels but not in cardiac muscle itself. **d** shows a pictorial representation of nodose and stellate ganglia, which are the major ganglia modulating sympathetic outflow to the heart (stellate ganglion) and parasympathetic sensory neurons (nodose ganglion). **e** shows silver staining of nodose ganglia cell isolated from D-alanine-fed (0.5 M D-alanine for 7 weeks) DAAO-TG^Cdh5 or control mice. Strongly positive silver staining is seen in the nodose ganglion from D-alanine-fed transgenic but not control mice. Scale bars, 100 μm. **f, g** show a series of photomicrographs and immunofluorescence images from isolated nodose ganglia (**f**) or stellate ganglia (**g**). The GFP stain for the transgene is strikingly positive in neurons of the nodose ganglia, but neurons in the stellate ganglia are negative for transgene protein expression; transgene expression in the representative stellate ganglion shown in (**g**) is restricted to GFP staining in a small blood vessel that can be seen coursing through the ganglion. Staining with antibodies directed against the sympathetic neuron marker tyrosine hydroxylase (TH) identify sparse nodose neurons, while most stellate neurons are positive for TH expression, as previously reported[69]. These images are representative of $n = 3$ independently-treated mice for each genotype. Scale bars, 100 μm.

pathology (Fig. 4), precisely mirroring the actual phenotype that we documented in these mice in vivo (Figs. 1, 2 and 5). Pathway enrichment analyses based on these proteomic data primarily shows enrichment of metabolic and immune pathways in the DRG proteome in D-alanine-treated DAAO-TG^Cdh5 mice, along with increases in related oxidative stress relevant pathways (ROS related pathways, HIF signaling, neurodegeneration and ferroptosis). The transcriptomic and proteomic analyses provide concordant evidence that chemogenetic oxidative stress leads to the activation of pathways related to adaptive and innate immune responses within the DRG. A small number of transcripts showed striking increase in abundance (Supplementary Fig. 3b). Tenascin C (*Tnc*), is expressed in extracellular matrix and has been involved in neuromuscular junction development and peripheral nervous system axon regeneration during disease states that involve inflammation or stress[39]. Betacellulin (*Btc*) has role in activation of EGFR signaling[40] and ATF3 promotes ferroptosis[41]. Taken together, the transcriptomic and proteomic signature of chemogenetic oxidative stress in DRG are consistent with an inflammatory neuropathic state that forms the basis for a striking sensory ataxia and an unexpected cardiac phenotype.

We were intrigued to discover that D-alanine-fed DAAO-TG^Cdh5 mice develop cardiac hypertrophy (Fig. 5). This might reflect the consequences of $H_2O_2$ generation either in the cardiac vasculature; in cardiac myocytes or non-muscle cardiac cells; or in the sensory neurons that show such robust functional transgene expression. We found no evidence for transgene expression in heart tissues except in cardiac blood vessels, which showed robust staining for the transgene (Fig. 5c and Supplementary Fig. 6e). In order to distinguish the relative importance of endothelial vs. neuronal oxidative stress in modulation of cardiac hypertrophy and neurodegeneration, we generated a second transgenic line that expresses DAAO under control of a distinct endothelial cell-specific promoter, Tie2, which also has been extensively characterized[30]. When DAAO-TG^Tie2 mice are fed D-alanine, we saw no ataxia whatsoever, and we found that there was no expression of the Tie2-driven transgene in DRG neurons (Supplementary Fig. 7) despite robust transgene expression of the transgene in mesenteric (Supplementary Fig. 6b, c) and cardiac (Supplementary Fig. 6e) blood vessels. Importantly, D-alanine feeding of DAAO-TG^Tie2 mice did not yield any change whatsoever in echocardiographic parameters: there is no cardiac phenotype when DAAO expression is restricted to the vascular endothelium. We therefore believe that it is the neuronal and not the endothelial expression of DAAO that is responsible for the cardiac hypertrophy and ataxia that is seen after the induction of chemogenetic oxidative stress in the DAAO-TG^Cdh5 mice. Both the DAAO-TG^Tie2 and DAAO-TG^Cdh5 mouse lines have expression of the DAAO transgene in vascular endothelial cells (Supplementary Fig. 6), but only the DAAO-TG^Cdh5 line has expression in DRG (Supplementary Fig. 7), and the DAAO-TG^Cdh5 line alone shows neurodegeneration and cardiac hypertrophy. It is possible that both neuronal and endothelial transgene expression is required for the cardiac and neural phenotypes, but endothelial expression alone is not sufficient: the DAAO-TG^Tie2 line is a critical control that provides confidence in our conclusion that DRG expression of the transgene is necessary for the ataxia and cardiac hypertrophy.

DRG are peripheral sensory neurons, and the connections between peripheral sensory neurons and internal organs ("enteroception") are under intense investigation[42]. Two of the major ganglia involved in cardiac innervation are the nodose (sensory parasympathetic) and stellate (efferent sympathetic) ganglia[27]. We therefore focused on these two ganglia to explore their potential role in modulation of cardiac hypertrophy. There was no transgene expression in stellate ganglia, whereas nodose ganglia showed robust transgene expression in nearly all neurons, associated with marked neurodegeneration following D-alanine feeding (Fig. 4). As we saw in the DRG, there is great selectivity in nodose vs. stellate ganglia, but within the nodose ganglion most if not all the neurons express the transgene. We speculate that the derangement of parasympathetic modulation of the heart as a consequence of nodose ganglia degeneration leads to unopposed sympathetic stimulation from the stellate ganglia; a role for sympathetic overactivity and reduced parasympathetic activity in adverse cardiac remodeling has been identified in many experimental models and disease states[27,43–45]. The mechanisms whereby sympathetic overactivity and diminished parasympathetic activity leads to cardiac pathology are incompletely understood[46].

The combination of sensory ataxia (Fig. 1), DRG degeneration (Fig. 2d), mitochondrial dysfunction (Fig. 2e), and cardiac hypertrophy (Fig. 5a, b) is similar to the constellation of findings found in patients with Friedreich's ataxia[47] the most common form of inherited ataxia in humans. The D-alanine-fed DAAO-TG^Cdh5 mouse may represent a potentially informative model of Friedreich's ataxia (FA), a progressive neurodegenerative disease in which death most commonly is a consequence of cardiac hypertrophy. FA is caused by loss of function mutations in the frataxin gene, which encodes the chaperone protein that is essential for assembly of iron-sulfur proteins in mitochondria[48]. The cardiac phenotype in FA patients is believed to be a consequence of frataxin deficiency in cardiac myocytes, which would then lead to mitochondrial dysfunction and cardiac dysfunction. Frataxin deficiency would lead to oxidative stress that initially involves different intracellular oxidants than are seen in our chemogenetic approach. In FA, the deficient assembly of iron-sulfur proteins leads to formation of oxygen radicals (superoxide radical—which is rapidly dismutated into hydrogen peroxide—as well as hydroxyl radical). By contrast, DAAO

directly generates hydrogen peroxide, and does not lead to the formation of oxygen radicals. But despite the differences in the specific ROS that are formed initially, the net cellular effect is oxidative distress due to the accumulation of excessive intracellular oxidants in neurons both in FA and in our chemogenetic approaches.

Our current findings suggest that the degeneration of sensory neuronal ganglia as a consequence of neuronal oxidative stress may also play an important pathophysiological role in the adverse cardiac remodeling that is seen in the hearts of FA patients. The transgenic/chemogenetic approach used in these studies provides an independent line of investigation that points to important connections between peripheral sensory nerves and cardiac remodeling, and may lead to insights into the molecular pathogenesis of Friedreich's ataxia.

## Methods

### Mouse models
All animal experiments were carried out under NIH guidelines for the care of laboratory mice, and all animal protocols were approved by the Brigham and Women's Hospital Institutional Animal Care and Use Committee (protocol 2016N000278). Mice were housed (five animals/cage maximum) in cages with unrestricted food (regular diet #5053) and drinking water access in a 12 h light–dark cycle. Room temperature was maintained at $21 \pm 2\,°C$ with 35% humidity. Equal numbers of male and female mice were studied, and studies were commenced when the animals were 8–12 weeks of age. A transgenic conditionally activatable HyPer-DAAO construct was made in collaboration with Novartis by cloning a stop codon flanked by loxP sites into the 5′-coding region of the cDNA encoding the cytosolic HyPer-DAAO fusion protein (the nucleotide sequence is in reference[7]) downstream of the CAG promoter (Supplementary Fig. 1). Using this approach, synthesis of HyPer-DAAO protein is blocked at the "floxed" stop codon until expression of Cre recombinase excises the stop codon and permits transcription of the full-length transgene. This construct was directed to the Rosa26 locus by CRISPR/Cas9 methods and transgenic founder lines were generated in C57/Bl6 mice using standard methods; founder lines were identified by PCR, confirming insertion of a single copy of the intact transgenic construct into the Rosa26 locus. This DAAO-TG[LoxP] mouse was then crossed with mice expressing Cre recombinase under control of the endothelial cell-specific Cdh5 promoter (strain 033055, Jackson Labs). DAAO-TG[Cdh5] positive offspring were identified by PCR, and are maintained in a C57/Bl6 background. Littermates containing Cre but lacking the transgene (Cre[+]/TG[−]) served as controls in order to control for possible off-target effects from Cre recombinase expression[49]. We generated a second transgenic mouse line DAAO-TG[Tie2] by crossing the DAAO-TG[LoxP] mouse with a commercially available mouse line (Jackson Labs strain 008863) that expresses Cre recombinase under control of the endothelial cell-specific Tie2 promoter[30]. A third mouse line was also generated in collaboration with Novartis in which the DAAO-HyPer transgene is under control of the cardiac-specific Myh6 promoter to yield the DAAO-TG[Car] line (Supplementary Fig. 5). All strains are maintained on the C57BL/6 background. The primer sequences used for genotyping are: Forward: TTCCCTCGTGATCTGCAACTC and reverse: CTTTAAGCCTGCCCAGAAGACT for Rosa26 wild-type; Forward: TTAATCCATATTGGCAGAACGAAAACG and reverse: CAGGCT AAGTGCCTTCTCTACA for recognition of Cre recombinase; and Forward: GGGAGGTGTGGGAGGTTTT and Reverse: CTTTAAGCCTGCCCA GAAGACT for detection of the HyPer-DAAO transgene. All physiological and imaging analyses were performed by personnel blinded to genotype and/or treatment. When mice become incapacitated by ataxia or showed signs of weight loss or distress, they were euthanized with isoflurane (5% in $O_2$) followed by decapitation.

### Histology and immunoblotting
Freshly isolated tissues from heart, spinal cord, nodose ganglia, stellate ganglia or dorsal root ganglia were fixed, embedded in paraffin, sectioned (10 μm), and slides were prepared and stained using hematoxylin/eosin stain or silver stain by the Rodent Histology Core at Harvard Medical School[50,51]. Images were acquired using a Axioskop microscope (ZEISS, Oberkochen, Germany) equipped with a Excelis MPX-20C Camera (Accu-Scope, Commack, NY, USA) and a Achroplan, ×10/0.25 Ph1 objective (ZEISS). Images were acquired with Capta Vision software (Accu-Scope).

For immunoblotting of cerebral artery, aorta, or mesenteric artery, blood vessels were isolated, mechanically dissociated and lysed in RIPA lysis buffer (Boston BioProducts, Boston, MA, USA) containing a protease/phosphatase inhibitor cocktail (Roche, Basel, Switzerland). After lysis, tissue samples were centrifuged at 12,000 rpm for 25 min. Protein concentration was quantified using the bicinchoninic acid protein assay (Thermo Fisher Scientific, Waltham, MA, USA), and equal amounts of protein (20 μg) were combined with 4× Laemmli buffer (Bio-Rad Laboratories, Hercules CA, USA), separated on 10% polyacrylamide gels (Bio-Rad Laboratories) and transferred onto nitrocellulose membranes (Bio-Rad Laboratories). Membranes were washed in TBST (Tris Buffered Saline with 0.1% Tween-20, Boston BioProducts) and blocked in TBST containing 5% (w/v) Blotting-Grade Blocker (Bio-Rad Laboratories) for 1 h. Membranes were incubated overnight in TBST containing 5% Blocker and primary antibodies for GFP or GAPDH (see antibodies section). Membranes were washed with TBST for $3 \times 10$ min and incubated for 2 h with a horseradish peroxidase-labeled goat anti-rabbit immunoglobulin (Cell Signaling Technology, Danvers, MA, USA) secondary antibody (1:1000 dilution) in TBST containing 5% Blocker. The membranes were washed 3 more times in TBST, loaded with a chemiluminescent reagent according to the manufacturer's protocols (SuperSignal West Femto, Thermo Fisher Scientific), and imaged using a ChemiDoc™ MP Imaging System (Bio-Rad Laboratories).

### Immunofluorescence and electron microscopy
For immunofluorescence of mouse tissues[52], mice were anesthetized using isoflurane followed by intracardiac perfusion with PBS/4% PFA followed by organ harvest, paraffin embedding, and tissue sectioning. Samples were treated with specific primary antibodies (see Antibodies section) conjugated with fluorescently-labeled secondary antibodies. Images were captured using an Olympus IX81 Microscope with an ImageEM CCD camera (Hamamatsu) and MetaMorph software (Molecular Devices).

For transmission electron microscopy[53], dorsal root ganglia (DRG) were isolated from euthanized mice and placed in EM fixative buffer (2.5% formaldehyde, 2.5% glutaraldehyde in 0.1 M sodium cacodylate buffer, pH 7.4), and then postfixed with 1% Osmium tetroxide ($OsO_4$)/1.5% Potassium ferrocyanide ($KFeCN_6$) for 1 h. Samples were washed twice with water, once with 50 mM maleate buffer (MB) pH 5.15, and then incubated in 1% uranyl acetate in MB for 1 h and washed with MB followed by 2 washes with water. The samples were then dehydrated in increasing concentrations of ethanol (10 min each in 50, 70, 90, and 100% ethanol), and then embedded in TAAB Epon (polymerized at 60 °C for 48 h) to cut ultrathin (60–80 nm) sections using a Reichert Ultracut-S microtome, and placed copper grids; images were taken using a Transmission Electron Microscope (JEOL 1200EX, Harvard EM Facility, Boston, MA, USA). All Images were captured with an AMT 2k CCD camera, and processed with FIJI software (National Institute of Health, Bethesda, MD, USA).

### Antibodies
The following primary antibodies were used: GFP (Cell Signaling Technology catalog number 2956, clone no. D5.1; dilution factor 1:1000,); Cdh5 (Santa Cruz Biotechnology catalog number sc9989, lot no. c1622, clone no. F-8; dilution factor 1:200), GAPDH (Cell Signaling Technology catalog number 2118, lot. No. 16, clone no. 14C10; dilution factor 1:2000,), NeuN (Millipore catalog number mab377, lot no.

382727; dilution factor 1:100), and Tyrosine Hydroxylase (EMD Millipore catalog number AB152, lot. No. 3845 [clone number is not available]; dilution factor 1:1000).

## Isolation of cardiomyocytes

Adult mouse cardiac myocytes were isolated from 8–12 week-old mice[54,55]. Mice were lightly anesthetized with isoflurane, heparinized (50 units, i.p.), and euthanized. The hearts were promptly removed from the chest and retrogradely perfused through the aorta with $Ca^{2+}$ free perfusion buffer. Hearts were digested with type 2 collagenase (Worthington, Lakewood, NJ, USA) at a concentration of 2.4 mg/ml. After 7–8 min of digestion, the heart was cut with sterile scissors just below the atria and minced with sterile fine forceps, and further dissociated by gentle agitation, followed by the stepwise introduction of $Ca^{2+}$ in myocyte stopping buffer, gradually increasing $Ca^{2+}$ to a final concentration of 1.2 mM. After $Ca^{2+}$ introduction was complete, the cardiomyocytes were pelleted, counted, and plated.

## Isolation and culture of dorsal root ganglia (DRG) neurons

To isolate and culture DRG[56,57], vertebral columns were dissected from euthanized mice, and and lumbar DRGs were isolated. The ganglia were digested with a 0.2 mg/ml collagenase type II solution (Worthington) for 45 min at 37 °C followed by a 20 min incubation after the addition of 2% trypsin. DRGs were mechanically dissociated using a 1 ml syringe by aspirating and ejecting the digested DRG solution ten times with decreasing needle size from 18 gauge to 28 gauge. The cell solution was centrifuged for 5 min at $160 \times g$, and DRG neurons were resuspended in Neurobasal complete medium supplemented with 5% horse serum and 50 ng/ml β-nerve growth factor (Thermo Fisher Scientific). The cells were then seeded on poly-L-lysine hydrobromide (Sigma-Aldrich, St. Louis, MO, USA) treated glass cover slides and incubated for 1–3 days in a humidified incubator (5% $CO_2$/95% air) at 37 °C before analysis.

## Live cell and tissue imaging

We performed real-time fluorescence measurements of HyPer from tissues or cells expressing HyPer-DAAO using an inverted Olympus IX81 microscope (Olympus, Waltham, MA, USA) with a ×10 air or a ×20 oil immersion objective (UPlanSApo, Olympus)[58]. Freshly isolated vascular tissue or live cultured DRG cells were seeded onto glass cover slides and placed in a perfusion chamber containing a Hepes-buffered salt solution containing (in mM):138 NaCl, 5 KCl, 2 $CaCl_2$, 1 $MgCl_2$, 10 D-glucose, and 10 HEPES, pH adjusted to 7.4. HyPer fluorescence was monitored at excitations of 420 or 490 nm and emission at 530 nm using optical filters (Semrock, Rochester, NY, USA). D-alanine or $H_2O_2$ were either added directly or administered through a gravity-based perfusion system during live fluorescence imaging experiments. Images were recorded at 10 s intervals with an ImagEM CCD camera (Hamamatsu, Bridgewater, NJ, USA) and data acquisition was processed with Metafluor Software (Molecular Devices, San Jose, CA, USA). Background readings were subtracted and the change in HyPer fluorescence ratio ($R = F_{490}/F_{420}$) was expressed as $R/R_0$, where $R$ is the ratio at time $t$, and $R_0$ is the baseline average ratio of 180 s at the start of experiment.

For measurement of mitochondrial membrane potential in DRG neurons using tetramethylrhodamine methyl ester (TMRM, Thermo Fisher Scientific), DRGs were isolated from D-alanine-fed (0.75 M for 5 days) DAAO-TG$^{Cdh5}$ or control mice, dissociated and seeded on μ slide 8 wells (ibidi, Martinsried, Germany). The cultured neurons were loaded with 3 nM TMRM[59] in Neurobasal complete medium (Thermo Fisher Scientific) at 37 °C for 30 min, and single-cell fluorescence was recorded with a ×20 oil immersion objective at 551 nm excitation and 576 nm emission using the Olympus IX81 microscope and Metafluor acquisition software. Single-cell TMRM fluorescence intensities were calculated after background subtraction.

## Mouse behavior analyses

Behavioral testing was performed in the Neurobehavioral Laboratory Core facility at Harvard NeuroDiscovery Center. Mice were housed at the facility for the duration of the testing protocol. Prior to all behavioral tests, mice were placed in the testing room to acclimatize for 30 min. The animals were monitored daily by the investigators and by the animal facility's staff. When mice developed ataxia to the point that they were unable to feed, they were euthanized with isoflurane (5% in $O_2$) followed by decapitation, consistent with the recommendations of the Panel on Euthanasia of the American Veterinary Medical Association. The Harvard Standing Committee on Animals oversees all procedures on animal including housing, feeding protocols, experimental procedures, and sacrifice of animals. The Institution also accepts as mandatory the PHS "Policy on Humane Care for the Utilization and Care of Vertebrate Animals Used in Testing, Research and Training". Open field test: Locomotor activity was measured using an automated open field test (OPT) monitoring system (Med Associates; St Albans, VT, USA) mounted with orthogonal infrared arrays[60]. Each mouse was placed into the 27 cm × 27 cm × 20.3 cm OPT box, which allowed them to move freely, and all movements were recorded over a 30-min testing period using Med Associates software (Activity Monitor, Version 5.9). Rotarod test: Rotarod analysis was performed[60] using an accelerating Rotarod apparatus (Stoelting; UgoBasile Apparatus). After animal training, animals were placed on the Rotarod and the latency to fall was recorded. Grip strength test: Grip strength was measured using the BIO-GS3 grip strength test apparatus (Bioseb) connected to a digital dynamometer[61].

## RNA sequencing of DRG

Immediately following animal sacrifice, dorsal root ganglia were dissected and snap-frozen in liquid nitrogen. RNA and cells were lysed by tissue fragmentation and mRNA was purified from total RNA using poly-T oligo-attached magnetic beads. First-strand cDNA was synthesized using random hexamer primers, followed by second strand cDNA synthesis using either dUTP for the directional library or dTTP for the non-directional library. Nucleotide sequencing was performed by Novogene (Durham, NC, USA). Gene Ontology (G.O.) analysis was performed on differentially expressed genes using DESeq2 and edgeR software. Differentially expressed genes were selected summing | $\log_2$FoldDifference| ≥ 1) and padj ≤ 0.05 between DAAO-TG$^{Cdh5}$ and control (Cdh5$^{Cre}$) littermate mice in their DRG neurons, as described[62,63]. Significantly differentially expressed genes were visualized using heatmaps and volcano plots, using ClusterProfiler software (R package)[64]. The list of genes that were significantly upregulated or downregulated in DAAO-TG$^{Cdh5}$ mice compared with littermate control was analyzed using the Single-Cell Portal (https://singlecell.broadinstitute.org/single_cell) and overlayed on the single-cell RNA sequencing data (https://singlecell.broadinstitute.org/single_cell/study/SCP1539/dorsal-root-ganglia-cells-after-nerve-injury) reported by Jager et al.[18]. The raw data and processed data for bulk RNA-seq of DRG neuronal cells were deposited in Genome Sequence Archive with accession ID GSE229143.

## DRG proteomic analysis

DRG proteomic analysis was performed[65] after DRG tissue was dissected from transgenic and control mice and homogenized and lysed with RIPA buffer. Then total protein was precipitated by TCA precipitation and resolubilized in RapiGestSF (Waters, Milford, MA, USA). Samples were reduced with 10 mM dithiothreitol (DTT) for 30 min at 80 °C, alkylated with 20 mM iodoacetamide (IAA) for 30 min at room temperature. And then 10 μl (20 ng/μl) of modified sequencing-grade trypsin (Promega, Madison, WI) was spiked into 300 μl PBS and the samples were placed in a 37 °C room overnight. Samples were acidified by adding 20 μl 20% formic acid solution and then desalted by STAGE tip (1). On the day of analysis, the samples were reconstituted in 10 μl of HPLC solvent A (0.1% formic acid). A nano-scale reverse-phase HPLC capillary column was created by packing 2.6 μm C18 spherical silica

beads into a fused silica capillary (100 μm inner diameter $x$ ~ 30 cm length) with a flame-drawn tip. After equilibrating the column each sample was loaded via a Famos auto sampler (LC Packings, San Francisco CA) onto the column. A gradient was generated, and peptides were eluted with increasing concentrations of solvent B (97.5% acetonitrile, 0.1% formic acid). After elution, the peptides were subjected to electrospray ionization into an LTQ Orbitrap Velos Elite ion-trap mass spectrometer (Thermo Fisher Scientific) to get detected, isolated, and fragmented to produce a tandem mass spectrum of specific fragment ions for each peptide.

Proteins were identified from raw data files using Proteome Discoverer (v 2.2) and searched against the SwissProt Mus musculus database using Sequest (Thermo Fisher Scientific) with the following search parameters: fixed modification, Carbamidomethyl (C); variable modifications, deamidation (NQ), oxidation (M); peptide mass tolerance 5 ppm; fragment mass tolerance 0.8 Da; enzyme, trypsin; max 2 missed cleavages. Proteins were selected based on a threshold range of 0 (min) to 3 (max) unique peptides during comparison with other samples. All databases include a revised version of all the sequences, and the data were filtered to 1–2% peptide false discovery rate. Annotated proteins were further analyzed by GeneCodis and Panther software packages to characterize pathway enrichment. The mass spectrometry proteomics data have been deposited to the ProteomeXchange Consortium via the PRIDE[66] partner repository with the dataset identifier PXD041382.

## Echocardiography

Echocardiography was performed using a Visual Sonics 2100 system equipped with a MS-550 probe[7]. The mice were initially anesthetized with 2–3% isoflurane and then titrated to 1–1.5% isoflurane to maintain heart rate above 450 bpm. Echocardiographic images of mid-papillary level short-axis and long-axis views of the heart were obtained. M-mode images were analyzed with Vevo LAB software (V.3.1.1 FUJI-FILM Visualsonics, Toronto, Canada). The sonographer and analyzer were blinded to the experimental treatment and/or genotype.

## Statistical analyses

Statistical analysis for in-between group comparisons was performed using Student's $t$ test (for two group comparisons) or two-way ANOVA with appropriate post test (for 3 group comparisons). Data values are presented as individual data points and expressed as means ± standard error of mean (SEM). Individual statistical tests are described in the corresponding figure legends. A $p$ value of <0.05 was considered statistically significant. Equal numbers of male and female mice were studied, and 6–8 animals of each sex were analyzed for each experimental treatment and genotype, except as noted in the figure legends. Sex was not studied as a biological variable because our initial observations indicated that both male and female DAAO-TG[Cdh5] transgenic mice develop ataxia after D-alanine feeding, and the effect(s) of sex on the phenotype were not further characterized in these studies. Data disaggregated for sex are provided in the source data file. All behavioral, physiological, and imaging studies performed and analyzed by scientists blinded to genotype and treatment. Statistical analyses were performed using GraphPad Prism 11.0 (GraphPad Software, La Jolla, CA).

## Reporting summary

Further information on research design is available in the Nature Portfolio Reporting Summary linked to this article.

## Data availability

Data supporting the findings of this study are available in the article and its Supplementary Information. Source Data file has also been deposited in figshare under accession code https://doi.org/10.6084/m9.figshare.22803020. The raw data and processed data for bulk RNA-seq of DRG neuronal cells were deposited in Genome Sequence Archive with accession ID GSE229143. The list of genes that were significantly upregulated or downregulated in DAAO-TG[Cdh5] mice compared with littermate control was analyzed using the Single-Cell Portal and overlayed on the single-cell RNA sequencing data (https://singlecell.broadinstitute.org/single_cell/study/SCP1539/dorsal-root-ganglia-cells-after-nerve-injury) reported by Jager et al.[18]. Raw data related to mouse genetic resources can be obtained from the NCBI Repository by requesting and following the guidelines for Genome Sequence Archive for noncommercial use GSE229143. The mass spectrometry proteomics data have been deposited to the ProteomeXchange Consortium via the PRIDE[66] partner repository with an accession ID PXD041382. Source data are provided with this paper.

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

## Acknowledgements

We thank Dr. Barbara Caldarone of the Harvard Medical School Mouse Behavior Core Lab for her assistance in the mouse behavioral studies. This work was supported by National Institutes of Health (NIH) grants R33 HL157918, R21 AG063073, and R01 HL152173 (to T.M.); NIH grant 5T32HL007609-34 (to F.S.) and by a FWF fellowship award to M.W.W. (J4466-B).

## Author contributions

S.Y., M.W.W., A.K.P., B.S., and T.M. designed experiments. S.Y., M.W.W., F.S., R.B., A.K.P., A.A.D., A.C.S., V.T. and T.A.C. performed and analyzed experiments. S.C., B.S., T.M. and W.C. designed and generated the DAAO-TGLoxP and DAAO-TGMyh6 mouse lines. T.M. supervised the project. S.Y. and T.M. wrote the manuscript.

## Competing interests

The authors declare no competing interests.
