## [Peer review file · Nature Communications]

REVIEWER COMMENTS

Reviewer #1 (Remarks to the Author):

This manuscript describes the generation of DAAO-TG Cdh5 transgenic mice express DAAO under control of the putatively endothelial-specific Cdh5 promoter, which turned out to be expressed in sensory DRG neurons. This led to an ataxic and cardiac phenotype with degeneration of sensory afferents and DRG neurons--however another transgenic line expressing DAAO under control of a different endothelial (this time truly specific) cell-specific Tie2 promoter did not show any neurological abnormalities arguing for a neuronal direct damage.

This is an interesting paper that is overall well executed and supported by the data.

However, I have a few questions:

It is not clear to me how the authors can distinguish the neuronal vs endothelial damage causing the observed phenotype: is the cdh5 promoter not expressed in endothelial cells around sensory DRG neurons or axons in the spinal cord and sciatic nerve? Is it not expressed in endothelial cells in the heart?

It appears that cdh5 is expressed in all DRG neurons: do the mice express other sensory disturbance, do they have alteration in pain perception?

How selective is the phenotype and the damage to specific neuronal subpopulation if at all?

Reviewer #2 (Remarks to the Author):

The phenotype of the mouse is very clear and this may have important implications for our understanding of sensory ataxia and the therefore our ability to treat it.

There is one major issue that currently limits the conclusion about the causative role of ROS / hydrogen peroxide in the pathogenesis of dysfunction described and this would be important to address. This issue relates to the fact that the dysfunction induced by the D-amino acid administration and ROS / hydrogen peroxide generation may not be the same mechanism that causes the disease. The dysfunction caused by elevating ROS / hydrogen peroxide may have little to do with the mechanisms causing the disease. It could be argued that any system forced to generate ROS / hydrogen peroxide will become dysfunctional. Can the authors at least somehow show the degrees of oxidative stress induced by the D-amino acid is approximately the same as the occurring in the real world disease? Are other models of ataxia protected by antioxidants?

Reviewer #3 (Remarks to the Author):

This manuscript describes an exceptionally rigorous experimental interrogation of the cell and functional impact of peroxide generation within specific cells in the brain and heart. I have absolutely no concerns about the facts that the authors establish. In parallel with the detailed experimental plan is the clarity of the writing; despite the density of the data, the manuscript was an 'easy' read, conveying the specifics of the experimental findings without elaboration and embellishment. This work and its presentation is of the highest caliber and I would strongly support its publication in Nat. Comm.

However, there is one aspect that needs to be attended to and that is the putative relationship this work has to Friedreich's Ataxia, or any ataxia for that matter. FRDA and essentially all genetic ataxias are 'mitochondrial' diseases and aside from 'oxidative stress' the cells are strongly compromised to deal with any stress. This is not true of the animal models here. Indeed, the one piece of evidence that the authors provide about "mitochondrial dysfunction" (line 311) is found in Fig.2e which shows alterations in mitochondrial morphology but offer no data about mitochondrial function. Nothing as simple as TMRM staining, or, better yet, Seahorse analysis. Without data like these, this paper is solely about the impact of peroxide stress and says nothing about any clinically evaluated ataxia.

Another issue with the authors overselling this work with respect to actual disease progression is their failure to consider if peroxide stress is a valid chemical model for physiologic processes. Peroxide itself is rather benign; the problem with it is its reaction with 'free' ferrous iron (Fenton chemistry) that generates the hydroxyl radical. First, the cell likely never is challenged by the levels of peroxide generated in the DAAO model and second, the cells in the model here don't have a defect in iron metabolism (as in FRDA, for example) that is a common comorbidity in mitochondrial energy metabolism associated neurodegenerative diseases. Specifically related to this DAAO model, brain iron accumulation is a clinical finding in Parkinson's Disease; PD and its MPTP, MAO backstory is obviously relevant to the work here in as much as the DAAO model follows from this aspect of the cell comorbidities found in PD.

Before the authors want to sell this system as a FRDA model they need to interrogate the iron and energy metabolism in it. Absent that work, they need to tone down their 'story' and tell one that pertains solely to the data at hand.

Reviewer #4 (Remarks to the Author):

Based on transgenic chemogenetic mouse lines that express yeast D-amino acid oxidase (DAAO) in endothelial cells and neurons, the work reported by Yadav et al. purportedly “confirmed that neurovascular oxidative stress is sufficient to cause sensory ataxia and cardiac hypertrophy, and identified the DAAO-TG^{Cdh5} mouse as a potentially informative animal model for Friedreich’s ataxia (FA)” (Lines 37-39). Several major caveats in terms of rationale, originality, study design, statistics, data interpretation and presentation have been identified.

Rationale

1. This is not a study designed to decipher the causal relationship between oxidative stress and sensory ataxia and cardiac hypertrophy, as implicated in the title of the manuscript. Instead, it is a report of auxiliary and unexpected (Lines 60, 203, 278) phenotypes of neurodegeneration, mitochondrial dysfunction and cardiac hypertrophy observed from transgenic/chemogenetic mouse models on induction of oxidative stress.

2. Using the term “neurovascular” in title and throughout text as a surrogate for neurons or endothelial cells is highly inappropriate and misleading. “Neurovascular unit” comprises endothelial cells, pericytes or vascular smooth muscle cells, glia and neurons, and functions to control blood-brain barrier permeability and cerebral blood flow.

Originality

1. The fundamental premise of this study is that “DAAO is quiescent when expressed in untreated mammalian cells, which typically contain only L-amino acids. When cells or animals expressing yeast DAAO are provided with D-alanine, the intracellular H_2O_2 generated by DAAO causes oxidative stress” (Lines 50-54). It follows that the most relevant information revealed in this study on sensory ataxia, cardiac hypertrophy or phenotypes associated with FA are based on responses to H_2O_2 induced in the DAAO-TG^{Cdh5} mouse model.

2. Engagement of H_2O_2 in the pathology of FA (neurodegeneration, mitochondrial dysfunction and cardiac hypertrophy), an inherited autosomal recessive disorder caused by severely reduced levels of frataxin, has been documented for decades (e.g. PNAS 2008;105:611-616; Antioxidants 2014;3:592-603). A deficit of this mitochondrial iron chaperone is consistent with the notion that toxic hydroxyl radical generated from H_2O_2 via iron-catalyzed Fenton chemistry at least partially underlies the pathology associated with this disease.

Study Design

1. Two concentrations of D-alanine in the drinking water and two associated time-windows were used in this study: 0.75M in conjunction with 4-5 days of observation (Figures 1, 2 and Extended Figure 2), 0.5M in conjunction with observations over 6 weeks (Figures 3-5 and Extended Figures 3-5). Without providing behavioral, morphological and imaging measurements comparable to those obtained during the short-term experiments, the RNA sequence analyses, gene ontology analyses and proteomic analyses performed on samples obtained from dorsal root ganglia (DRG) during the long-term experiments are basically ineffectual in scientific sense with reference to FA.

2. Likewise, results from measuring echocardiographic parameters only during the long-term experiments dissociate them from the behavioral, morphological and imaging measurements obtained during the short-term experiments.

Statistics

1. One-way ANOVA is inappropriate for continuous data shown in Figures 1g, 1h, and Extended Figure 2b. Also, significance cannot be denoted for differences in individual means based on analysis of group means.

2. It is unclear from the figure legends as to how “equal numbers of male and female mice were studied, and 6-8 animals of each sex were analyzed for each experimental treatment and genotype” (Lines 834-836) were executed. In particular, how can this statement reconcile with the presentation of many results as “representative of n = or > 3 mice per group” (Lines 527-528, 540, 552).

Data Interpretation

Major flaws in data presentation, analysis and interpretation fail to support the claim that neurovascular oxidative stress is sufficient to cause sensory ataxia and cardiac hypertrophy, and the DAAO-TG^{Cdh5} mouse is a potentially informative animal model for FA.

1. Figure 1, Supplementary video and Extended Figure 2

a. The box in Figure 1a is erroneously labelled (see b. below).

b. Lines 100-101: The statement “the dorsal tracts of the spinal cord, which transmit sensory signals, had degenerated” is incorrect for at least two reasons. First, tracts are made up of fibers (white matter). The areas denoted by the arrows are located in the dorsal horn, which is made up of neurons (grey matter) and do not denote “tracts”. Second, and most importantly, conscious proprioception (sense of position) is transmitted via the dorsal column, which is the white matter located on both sides of the dorsal median sulcus of the spinal cord. The clinical sign of proprioception dysfunction is ataxia (incoordination). Since the dorsal column appears normal in Figures 1b and 1c, it is unlikely that ataxia has taken place.

c. Lines 101-103: Crucial data in support of the statement “the ventral (motor) tracts were apparently unaffected (Figure 1a-1c); peripheral nerves in the hindlimb skeletal muscle also showed signs of degeneration, while the skeletal muscle itself was normal” are missing. The photomicrographs of lumbar spinal cord (Figures 1b and 1c) do not even contain the ventral horn where the motor neurons are located.

d. What is the evidence that rules out the possibility that failure in behavioral tests exhibited by the DAAO-TG^{Cdh5} mice is because of a loss of muscle strength? Both the Supplementary video and Extended Figure 2a only showed locomotor activity of the mice despite the initial ataxia in the hindlimbs. Comparable data during the time when animals failed the behavioral tests are desirable.

e. Extended Figure 2b: It appears that results from treatment with antioxidants are still significantly different from control. Treatment with selective inhibitor of H₂O₂ may produce more convincing results.

2. Figure 2

a. Extensive network of blood vessels that encapsulate and encircle the cell body of sensory neurons in the DRG (Molecular Pain 2008; 4:10). Since the DRG is the key experimental target in this study, demonstration of colocalization of VE-cadherin (endothelial cell marker) or GFP (detects the YFP contained in the HyPer-DAAO fusion protein transgene) in the cell body-rich area of the DRG is a much better choice. Several major blood vessels to the brain is grouped under “cerebral artery”. Without specifying the particular vessel in (a), the term “cerebral artery” is meaningless anatomically.

b. Data from control mice are required to justify the presence of neuronal death (d) and swollen and distorted mitochondria (e).

c. More importantly, no experimental evidence is presented to support that H₂O₂ presumably induced in the DRG (c) is responsible for the results implicated in (d) and (e).

3. Figures 3, 4 and Extended Figures 3-5

a. It is noted that the authors have not provided any experimental data showing the extent of ataxia, nor corresponding changes in behavioral, morphological and imaging measurements 6 weeks after feeding DAAO-TG^{Cdh5} mice with 0.5M D-alanine. It follows that the myriad findings from RNA sequence analyses, gene ontology analyses and proteomic analyses may simply represent changes in transcriptomic and proteomic profiles in the DRG on chronic induction of H₂O₂. Nevertheless, there is no demonstration of even an association between those changes, ataxia and neurodegeneration in the DRG, let alone causation.

b. The mention of “muscle pathology” (Line 162) reinforces the possibility that the failure in behavioral tests exhibited by the DAAO-TG^{Cdh5} mice is because of a loss of muscle strength. It also contradicts the claim that “skeletal muscle itself was normal” (Line 103).

4. Figure 5 and Extended Figures 6, 7

a. Whereas findings from echocardiography suggest the development of cardiac hypertrophy in DAAO-TGCdh5 mice following chronic D-alanine feeding (Figure 5a,b), the values presented are within normal physiological ranges of the measured parameters.

b. The findings that the transgene is only expressed in cardiac blood vessels but not in heart tissues in DAAO-TGCdh5 mice that exhibited presumed cardiac hypertrophy contradict observations from the DAAO-TGTie2 mice and the statement “it is not the endothelial expression of DAAO that is responsible for the cardiac hypertrophy that is seen after the induction of chemogenetic oxidative stress in the DAAO-TGCdh5 mice (Lines 294-296).

c. The suggestion that it is the neuronal expression of DAAO that is responsible for the cardiac hypertrophy (Lines 294-295) is also elusive. There is no mention as to the location of the neurons that express DAAO, and how H2O₂ induced in these neurons elicit cardiac hypertrophy.

d. Reduced parasympathetic activity in the statement “a role for sympathetic overactivity and reduced parasympathetic activity in adverse cardiac remodeling” (Lines 306-307) refers to reduced efferent vagal influence to the heart that originates from the nucleus ambiguus. It is not the same as “loss of parasympathetic modulation of the heart as a consequence of nodose ganglia degeneration (Lines 304-305).

Presentation

1. Line 24: “D-amino oxidase” should read “D-amino acid oxidase”.

2. “Friedreich’s Ataxia” should read “Friedreich’s ataxia”; and should be abbreviated as “FA” only when first appeared in text (Line 37) and thereafter (Lines 39, 164, 170, 312, 314, 315).

3. Ref #8: No information is provided on journal, volume and pagination.

4. Extended Figure 2: “b” above “DAAO-TGCdh5 mice” should be deleted.

Reviewer #1 (Remarks to the Author):

This manuscript describes the generation of DAAO-TG Cdh5 transgenic mice express DAAO under control of the putatively endothelial-specific Cdh5 promoter, which turned out to be expressed in sensory DRG neurons. This led to an ataxic and cardiac phenotype with degeneration of sensory afferents and DRG neurons--however another transgenic line expressing DAAO under control of a different endothelial (this time truly specific) cell-specific Tie2 promoter did not show any neurological abnormalities arguing for a neuronal direct damage.

This is an interesting paper that is overall well executed and supported by the data. However, I have a few questions:

It is not clear to me how the authors can distinguish the neuronal vs endothelial damage causing the observed phenotype: is the cdh5 promoter not expressed in endothelial cells around sensory DRG neurons or axons in the spinal cord and sciatic nerve? Is it not expressed in endothelial cells in the heart?

Our response: We thank the reviewer for their positive comments.

Both the Cdh5 and the Tie2 promoter drive expression in vascular endothelial cells throughout the body (ref. 35), including DRG neurons and heart. **The difference lies in the fact that only the Cdh5 promoter is active in sensory neurons, whereas the Tie2 transgenic mouse shows no expression whatsoever in neurons (see Extended Data Figure 7).** It is the fact that the transgenic Cdh5 mice have ataxia and the Tie2 mice do not-- while both lines have endothelial transgene expression-- that gives us confidence that **it is the additional and unique expression of Cdh5 in neurons that causes both the ataxia and the cardiac phenotype.** Yet we cannot exclude that the combination of endothelial *plus* neuronal expression is required for neurodegeneration, since the Cdh5 line has expression in both tissues. *But expression in endothelium alone is not sufficient for this phenotype since the Tie2 line has no neurodegeneration.* The revised manuscript explicitly addresses these points in the Discussion section to clarify this interesting query raised by the Reviewer.

It appears that cdh5 is expressed in all DRG neurons: do the mice express other sensory disturbance, do they have alteration in pain perception?

Our response: Yes, the Cdh5-driven transgene appears to be expressed in all or nearly all DRG neurons (Figure 2c), but behavioral analyses of sensation (e.g. pain, touch) in the treated mice are difficult to interpret because of the animals' profound ataxia. We now discuss this point in the Discussion section of the revised manuscript and thank the reviewer for this query.

How selective is the phenotype and the damage to specific neuronal subpopulation if at all?

Our response: The phenotype is highly selective for sensory neurons, but within the DRG most if not all the neurons express the Cdh5 transgene and there does not appear to be selectivity for specific neurons within the DRG. Furthermore, the nodose ganglion shows Cdh5 expression and degeneration in nearly all neurons (Figure 5f), yet the stellate ganglion is completely spared (Figure 5g). So the selectivity for sensory neurons is quite strict, but *within* the sensory ganglia all or nearly neurons appear to be affected. We now discuss this point in the Discussion section of the revised manuscript and appreciate the Reviewer raising this query.

Reviewer #2 (Remarks to the Author):

The phenotype of the mouse is very clear and this may have important implications for our understanding of sensory ataxia and the therefore our ability to treat it.

There is one major issue that currently limits the conclusion about the causative role of ROS / hydrogen peroxide in the pathogenesis of dysfunction described and this would be important to address. This issue relates to the fact that the dysfunction induced by the D-amino acid administration and ROS / hydrogen peroxide generation may not be the same mechanism that causes the disease. The dysfunction caused by elevating ROS / hydrogen peroxide may have little to do with the mechanisms causing the disease. It could be argued that any system forced to generate ROS / hydrogen peroxide will become dysfunctional. Can the authors at least somehow show the degrees of oxidative stress induced by the D-amino acid is approximately the same as the occurring in the real world disease? Are other models of ataxia protected by antioxidants?

Our response: We thank the Reviewer for their positive comments.

We totally agree with the reviewer that the specific intracellular oxidants that are initially generated by frataxin deficiency in FA are different than those that result from DAAO-generated H₂O₂ synthesis- *but in both cases the ultimate cellular outcome is oxidative stress, i.e. a net increase in intracellular oxidants.* The oxidative stress caused by frataxin deficiency in FA would initially involve different intracellular oxidants than are seen in our chemogenetic approach. In FA, the deficient assembly of iron-sulfur proteins (as a consequence of decreased frataxin abundance) leads to the formation of oxygen radicals (specifically, superoxide radical– which is rapidly dismuted into hydrogen peroxide– as well as hydroxyl radical). By contrast, DAAO directly generates hydrogen peroxide, and does not lead to the formation of these oxygen radicals. But despite the differences in the specific ROS that are formed initially, *in both cases the net cellular effect is oxidative distress* due to the accumulation of excessive intracellular oxidants in neurons both in FA and also in our chemogenetic approach. The overall redox balance of the cell reflects the integrated metabolism of many different cellular oxidants, and similar levels of oxidative stress may be initiated by distinct proximal pathways. We now discuss this very interesting point explicitly in the Discussion section of the revised manuscript and appreciate the reviewer raising this important topic.

In response to the Reviewer's next comment, we now cite papers showing that the increase in H₂O₂ following DAAO activation is within the range of oxidative stress seen in disease states, and have added literature citations to this effect (references 13, 40-42). The efficacy of antioxidants in treatment of FA is controversial, as are the broader effects of antioxidants in other disease states associated with oxidative stress (13)– perhaps reflecting the importance of intracellular oxidants in normal physiology. We now discuss these points in the revised ms.

Reviewer #3 (Remarks to the Author):

This manuscript describes an exceptionally rigorous experimental interrogation of the cell and functional impact of peroxide generation within specific cells in the brain and heart. I have absolutely no concerns about the facts that the authors establish. In parallel with the detailed experimental plan is the clarity of the writing; despite the density of the data, the manuscript was an 'easy' read, conveying the specifics of the experimental findings without elaboration and embellishment. This work and its presentation is of the highest caliber and I would strongly support its publication in Nat. Comm.

However, there is one aspect that needs to be attended to and that is the putative relationship

this work has to Friedreich's Ataxia, or any ataxia for that matter. FRDA and essentially all genetic ataxias are 'mitochondrial' diseases and aside from 'oxidative stress' the cells are strongly compromised to deal with any stress. This is not true of the animal models here. Indeed, the one piece of evidence that the authors provide about "mitochondrial dysfunction" (line 311) is found in Fig.2e which shows alterations in mitochondrial morphology but offer no data about mitochondrial function. Nothing as simple as TMRM staining, or, better yet, Seahorse analysis. Without data like these, this paper is solely about the impact of peroxide stress and says nothing about any clinically evaluated ataxia.

We thank the reviewer for their positive comments.

In response to the Reviewer's suggestion, we have performed TMRM staining in DRG isolated from D-alanine-fed DAAO-TG^{Cdh5} mice (with littermate controls). These studies indeed revealed decreased mitochondrial potential in DRG isolated from D-alanine-fed DAAO-TG^{Cdh5} mice, consistent with mitochondrial dysfunction. These data are included in the revised manuscript as a new Extended Data Figure 2c and are discussed in the manuscript.

Another issue with the authors overselling this work with respect to actual disease progression is their failure to consider if peroxide stress is a valid chemical model for physiologic processes. Peroxide itself is rather benign; the problem with it is its reaction with 'free' ferrous iron (Fenton chemistry) that generates the hydroxyl radical. First, the cell likely never is challenged by the levels of peroxide generated in the DAAO model and second, the cells in the model here don't have a defect in iron metabolism (as in FRDA, for example) that is a common comorbidity in mitochondrial energy metabolism associated neurodegenerative diseases. Specifically related to this DAAO model, brain iron accumulation is a clinical finding in Parkinson's Disease; PD and its MPTP, MAO backstory is obviously relevant to the work here in as much as the DAAO model follows from this aspect of the cell comorbidities found in PD.

Before the authors want to sell this system as a FRDA model they need to interrogate the iron and energy metabolism in it. Absent that work, they need to tone down their 'story' and tell one that pertains solely to the data at hand.

We apologize for giving the impression that we are "overselling this work" as an FRDA model: we just want to point out the striking similarities in the phenotype between our transgenic mouse and FRDA. We certainly take the Reviewer's important point that frataxin deficiency in FRDA and H2O2 generation as in the present study may alter cellular redox balance by different molecular mechanisms (see response to Reviewer 1), and we now discuss these points explicitly in the revised manuscript. We have changed our phrase stating that our transgenic/chemogenetic approach "could lead to an informative new model of Friedreich's ataxia" to now instead state that this approach could lead to "new insights into the molecular pathogenesis of Friedreich's ataxia". We appreciate the reviewer's important comments on this point, which is explicitly discussed in the revised manuscript.

Reviewer #4 (Remarks to the Author):

Based on transgenic chemogenetic mouse lines that express yeast D-amino acid oxidase (DAAO) in endothelial cells and neurons, the work reported by Yadav et al. purportedly "confirmed that neurovascular oxidative stress is sufficient to cause sensory ataxia and cardiac hypertrophy, and identified the DAAO-TGCdh5 mouse as a potentially informative animal model for Friedreich's ataxia (FA)" (Lines 37-39). Several major caveats in terms of rationale, originality, study design, statistics, data interpretation and presentation have been identified.

Our response: We appreciate the Reviewer's comprehensive critique and have tried our best to respond to the Reviewer's far-ranging comments and to clarify several of the concerns raised.

Rationale

1. This is not a study designed to decipher the causal relationship between oxidative stress and sensory ataxia and cardiac hypertrophy, as implicated in the title of the manuscript. Instead, it is a report of auxiliary and unexpected (Lines 60, 203, 278) phenotypes of neurodegeneration, mitochondrial dysfunction and cardiac hypertrophy observed from transgenic/chemogenetic mouse models on induction of oxidative stress.

Our response: The Reviewer is correct that our initial finding was unexpected, yet we believe that our somewhat serendipitous observation nevertheless led to some interesting new insights.

2. Using the term "neurovascular" in title and throughout text as a surrogate for neurons or endothelial cells is highly inappropriate and misleading. "Neurovascular unit" comprises endothelial cells, pericytes or vascular smooth muscle cells, glia and neurons, and functions to control blood-brain barrier permeability and cerebral blood flow.

Our response: We regret that the Reviewer felt that the term "neurovascular" in this paper is "inappropriate and misleading". A search in PubMed reveals that this term ("neurovascular" without the term "unit") is used in more than 23,000 published papers, so at least we are not alone in this usage. We have explained our definition of this term in the Introduction section of the revised.

Originality

1. The fundamental premise of this study is that "DAAO is quiescent when expressed in untreated mammalian cells, which typically contain only L-amino acids. When cells or animals expressing yeast DAAO are provided with D-alanine, the intracellular H₂O₂ generated by DAAO causes oxidative stress" (Lines 50-54). It follows that the most relevant information revealed in this study on sensory ataxia, cardiac hypertrophy or phenotypes associated with FA are based on responses to H₂O₂ induced in the DAAO-TGCdh5 mouse model.

Our response: We take the Reviewer's point but note that our transgenic/chemogenetic approach has never been used before to study oxidative stress in neurons in vivo. We feel that our model is novel and informative and permits the dynamic and highly specific regulation of oxidative stress in target tissues in vivo- something that is not possible with other approaches.

2. Engagement of H₂O₂ in the pathology of FA (neurodegeneration, mitochondrial dysfunction and cardiac hypertrophy), an inherited autosomal recessive disorder caused by severely reduced levels of frataxin, has been documented for decades (e.g. PNAS 2008;105:611-616; Antioxidants 2014;3:592-603). A deficit of this mitochondrial iron chaperone is consistent with the notion that toxic hydroxyl radical generated from H₂O₂ via iron-catalyzed Fenton chemistry at least partially underlies the pathology associated with this disease.

Our response: We agree with the Reviewer that the intracellular oxidants initially generated by frataxin deficiency are different than those are generated by DAAO (see our detailed response to Reviewer 1)- yet the ultimate cellular consequences converge on some similar phenotypes- reflecting the highly coordinated and complex intracellular pathways of redox metabolism. We now discuss this point explicitly in the revised manuscript.

Study Design

1. Two concentrations of D-alanine in the drinking water and two associated time-windows were used in this study: 0.75M in conjunction with 4-5 days of observation (Figures 1, 2 and Extended Figure 2), 0.5M in conjunction with observations over 6 weeks (Figures 3-5 and Extended Figures 3-5). Without providing behavioral, morphological and imaging measurements comparable to those obtained during the short-term experiments, the RNA sequence analyses, gene ontology analyses and proteomic analyses performed on samples obtained from dorsal root ganglia (DRG) during the long-term experiments are basically ineffectual in scientific sense with reference to FA.

Our response: We now state explicitly that after 6 weeks of lower-dose D-alanine feeding, the mice show similar behavioral, morphological and imaging characteristics as were seen after short-term treatments with higher-dose D-alanine. We thank the Reviewer for raising this query.

2. Likewise, results from measuring echocardiographic parameters only during the long-term experiments dissociate them from the behavioral, morphological and imaging measurements obtained during the short-term experiments.

Our response: The rationale behind performing echocardiographic analyses only after the long-term experiments is that the very rapid onset of ataxia with the high-dose D-alanine feeding seemed unlikely to provide sufficient time for cardiac remodeling to take place. Indeed, patients with FA often develop ataxia as children but pathological cardiac remodeling is more commonly seen in young adults. We clarify this point in the revised ms.

Statistics

1. One-way ANOVA is inappropriate for continuous data shown in Figures 1g, 1h, and Extended Figure 2b. Also, significance cannot be denoted for differences in individual means based on analysis of group means.

Our response: We regret that we made an error here: we had in fact performed two-way ANOVA for these figures and have corrected our error. We are grateful to the reviewer for noting this.

2. It is unclear from the figure legends as to how “equal numbers of male and female mice were studied, and 6-8 animals of each sex were analyzed for each experimental treatment and genotype” (Lines 834-836) were executed. In particular, how can this statement reconcile with the presentation of many results as “representative of n = or > 3 mice per group” (Lines 527-528, 540, 552).

Our response: We have now note in individual figure legends when the results presented reflect fewer than the 6-8 animals that were analyzed for most of the experiments shown, and we clarify this point in the Statistics section.

Data Interpretation

Major flaws in data presentation, analysis and interpretation fail to support the claim that neurovascular oxidative stress is sufficient to cause sensory ataxia and cardiac hypertrophy, and the DAAO-TGCdh5 mouse is a potentially informative animal model for FA.

Our response: we are frankly puzzled by the Reviewer’s statement, and regret that they have reached this conclusion— with which we must respectfully disagree. We did not pursue these

studies seeking to create a new animal model for FA, but we do feel that is valid to point out that several phenotypic features of our new chemogenetic model are also found in FA.

1. Figure 1, Supplementary video and Extended Figure 2

a. The box in Figure 1a is erroneously labelled (see b. below).

Our response: we have corrected the labeling in this Figure and regret our error.

b. Lines 100-101: The statement “the dorsal tracts of the spinal cord, which transmit sensory signals, had degenerated” is incorrect for at least two reasons. First, tracts are made up of fibers (white matter). The areas denoted by the arrows are located in the dorsal horn, which is made up of neurons (grey matter) and do not denote “tracts”. Second, and most importantly, conscious proprioception (sense of position) is transmitted via the dorsal column, which is the white matter located on both sides of the dorsal median sulcus of the spinal cord. The clinical sign of proprioception dysfunction is ataxia (incoordination). Since the dorsal column appears normal in Figures 1b and 1c, it is unlikely that ataxia has taken place.

Our response: We regret any misinterpretation of our results, and now show in Extended Data Figure 1c the full transverse sections of the lumbar spinal cord for D-alanine-treated control and DAAO-TG^{Cdh5} transgenic mice, over which we have overlaid a schematic of the major structural elements of the spinal cord, labeling the dorsal and ventral horn and other major elements. It can be seen that the areas noted by the arrows are indeed in the dorsal tracts, *not* in the dorsal horn, as suggested by the Reviewer. We hope that showing the entire transverse spinal cord section (along with the schematic) provides further support for our data showing that the dorsal tracts are affected in the D-alanine-fed DAAO-TG^{Cdh5} mice.

c. Lines 101-103: Crucial data in support of the statement “the ventral (motor) tracts were apparently unaffected (Figure 1a-1c); peripheral nerves in the hindlimb skeletal muscle also showed signs of degeneration, while the skeletal muscle itself was normal” are missing. The photomicrographs of lumbar spinal cord (Figures 1b and 1c) do not even contain the ventral horn where the motor neurons are located.

Our response: we showed a close-up of the dorsal tracts in Figure 1 to provide a clearer view of the regions in which we found pathology. We show in a new Extended Data Figure 1C a full transverse section of the lumbar spinal cord following D-alanine treatment of control and transgenic mice. As shown in this figure, the dorsal column shows vacuoles in the transgenic but not control mouse, and the ventral horn is normal in both images.

d. What is the evidence that rules out the possibility that failure in behavioral tests exhibited by the DAAO-TG^{Cdh5} mice is because of a loss of muscle strength? Both the Supplementary video and Extended Figure 2a only showed locomotor activity of the mice despite the initial ataxia in the hindlimbs. Comparable data during the time when animals failed the behavioral tests are desirable.

Our response: Our data do not support a mechanism for loss of muscle strength in the ataxia phenotype, and we discuss this point in the ms.

e. Extended Figure 2b: It appears that results from treatment with antioxidants are still significantly different from control. Treatment with selective inhibitor of H₂O₂ may produce more convincing results.

Our response: As we noted in the manuscript, the antioxidant treatment attenuated the ataxia phenotype observed in the animals that were treated D-alanine without antioxidants. The fact that antioxidants do not fully rescue the phenotype does not detract from our conclusion that the antioxidant treatment lessened the severity of the phenotype. As to the Reviewer's second point, we are unaware of the existence of any "selective inhibitors of H₂O₂".

2. Figure 2

a. Extensive network of blood vessels that encapsulate and encircle the cell body of sensory neurons in the DRG (Molecular Pain 2008; 4:10). Since the DRG is the key experimental target in this study, demonstration of colocalization of VE-cadherin (endothelial cell marker) or GFP (detects the YFP contained in the HyPer-DAAO fusion protein transgene) in the cell body-rich area of the DRG is a much better choice. Several major blood vessels to the brain is grouped under "cerebral artery". Without specifying the particular vessel in (a), the term "cerebral artery" is meaningless anatomically.

Our response: we now specify that the artery in question is the middle cerebral artery.

b. Data from control mice are required to justify the presence of neuronal death (d) and swollen and distorted mitochondria (e).

Our response: we now show EM images of DRG from control mice in a new Figure 2f, which reveal normal mitochondria.

c. More importantly, no experimental evidence is presented to support that H₂O₂ presumably induced in the DRG (c) is responsible for the results implicated in (d) and (e).

Our response: We frankly do not understand this statement, and regret that we were not to effectively communicate our experimental approach here. The fact that both D-alanine treatment *plus* transgene expression in DRG are required to produce the phenotype is emphasized throughout the manuscript. The lack of an ataxia or cardiac phenotype in the DAAO-TGTie2 mice- which exhibit endothelial but not DRG transgene expression- serves as a key control. We have expanded our discussion in attempt to clarify these points.

3. Figures 3, 4 and Extended Figures 3-5

a. It is noted that the authors have not provided any experimental data showing the extent of ataxia, nor corresponding changes in behavioral, morphological and imaging measurements 6 weeks after feeding DAAO-TGCdh5 mice with 0.5M D-alanine. It follows that the myriad findings from RNA sequence analyses, gene ontology analyses and proteomic analyses may simply represent changes in transcriptomic and proteomic profiles in the DRG on chronic induction of H₂O₂. Nevertheless, there is no demonstration of even an association between those changes, ataxia and neurodegeneration in the DRG, lest causation.

Our response: We now explicitly state that the longer-term treatment of DAAO-TG^{Cdh5} mice recapitulates the behavioral, morphological and imaging characteristics of the mice treated with higher D-alanine doses for a shorter period of time. We feel that our rigorous control experiments fully support our contention of causality and have expanded our discussion of this point in the revised ms.

b. The mention of "muscle pathology" (Line 162) reinforces the possibility that the failure in

behavioral tests exhibited by the DAAO-TGCdh5 mice is because of a loss of muscle strength. It also contradicts the claim that “skeletal muscle itself was normal” (Line 103).

Our response: there is no apparent muscle pathology in these mice. We have stated this point in the manuscript.

4. Figure 5 and Extended Figures 6, 7

a. Whereas findings from echocardiography suggest the development of cardiac hypertrophy in DAAO-TGCdh5 mice following chronic D-alanine feeding (Figure 5a,b), the values presented are within normal physiological ranges of the measured parameters.

Our response: We respectfully disagree with the reviewer. The appropriate comparator is *not* the “physiological range” of the echocardiographic parameters— these values that are influenced by animal age and by mouse strain and many other factors— but by the rigorous comparison with littermates that are negative for the transgene but express the Cre recombinase, which we have studied here at the same time and under the same conditions as the transgenic littermates. Transgene-negative littermates are the proper controls, not some literature-based “physiological range”, which would be inappropriate to use here.

b. The findings that the transgene is only expressed in cardiac blood vessels but not in heart tissues in DAAO-TGCdh5 mice that exhibited presumed cardiac hypertrophy contradict observations from the DAAO-TGTie2 mice and the statement “it is not the endothelial expression of DAAO that is responsible for the cardiac hypertrophy that is seen after the induction of chemogenetic oxidative stress in the DAAO-TGCdh5 mice (Lines 294-296).

Our response: Once again, we respectfully disagree with the Reviewer’s statement, and regret any miscommunication here. Both the DAAO-TG^{Cdh5} and the DAAO-TG^{Tie2} mice show transgene expression in the vascular endothelium— and neither line shows any transgene expression in cardiac myocytes. Only the DAAO-TG^{Cdh5} line shows transgene expression in sensory neurons, and only this transgenic line shows ataxia and cardiac hypertrophy. We have modified the manuscript to clarify this point.

c. The suggestion that it is the neuronal expression of DAAO that is responsible for the cardiac hypertrophy (Lines 294-295) is also elusive. There is no mention as to the location of the neurons that express DAAO, and how H₂O₂ induced in these neurons elicit cardiac hypertrophy.

Our response: We agree with the reviewer that mechanisms whereby DAAO expression in neurons leads to cardiac hypertrophy has not been established in these studies. This is an interesting and important area of future investigation, as was noted in the manuscript.

d. Reduced parasympathetic activity in the statement “a role for sympathetic overactivity and reduced parasympathetic activity in adverse cardiac remodeling” (Lines 306-307) refers to reduced efferent vagal influence to the heart that originates from the nucleus ambiguus. It is not the same as “loss of parasympathetic modulation of the heart as a consequence of nodose ganglia degeneration (Lines 304-305).

Our response: We have clarified these sentences in the revised manuscript and regret any miscommunication.

Presentation

1. Line 24: “D-amino oxidase” should read “D-amino acid oxidase”.

Our response: we have corrected this error.

2. “Friedreich’s Ataxia” should read “Friedreich’s ataxia”; and should be abbreviated as “FA” only when first appeared in text (Line 37) and thereafter (Lines 39, 164, 170, 312, 314, 315).

Our response: we have amended the text as suggested.

3. Ref #8: No information is provided on journal, volume and pagination.

Our response: this is a reference to a book that was published on line, and the formatting was inadvertently omitted by the EndNote bibliographic program that we used. We regret the error and thank the reviewer for pointing out the incomplete citation, which has been corrected in the revised ms.

4. Extended Figure 2: “b” above “DAAO-TGCdh5 mice” should be deleted.

Our response: we have corrected this error.

REVIEWER COMMENTS

Reviewer #1 (Remarks to the Author):

The authors have addressed my points.

Thank you for the nice report.

Reviewer #2 (Remarks to the Author):

The authors have dealt with my question about how the amount of peroxide generated by DAO used to cause the ataxia relates to the molecular species and their abundance in the the real world disease situation by adding some helpful Discussion text that reasonably addresses this issue. I think there is not much that can be done to fully tackle this question experimentally, and so I find the added text sufficient in the circumstances.

Reviewer #3 (Remarks to the Author):

I am impressed with the thoroughness with which the authors responded to comments from all reviewers. As I noted previously, I considered this work is of the highest caliber and eminently suitable for publication in Nat Comm. They have addressed my concerns, adding relevant data and altering their discussion to focus on the specific outcomes of their experiments.

Reviewer #4 (Remarks to the Author):

The authors have responded satisfactorily to the comments of this reviewer. To rephrase the notion that the transgenic/chemogenetic approach could lead to "an informative new model of Friedreich's ataxia" to "new insights into the molecular pathogenesis of Friedreich's ataxia" has resolved the critical concerns on this approach,.

Two minor suggestions:

1. Lines 108-109 and 401: To conform with Panels c and d in Extended Figure 1, it is recommended that “dorsal column” and “ventral horn” be used instead of “dorsal tracts” and “ventral (motor) tracts”.

2. Figure 1a: Note that “dorsal horn” is NOT part of the “Dorsal Column” and should not be included in the “box”; the term “Ventral Column” is anatomically ambiguous. Again, to conform with Panels c and d in Extended Figure 1, it is recommended that only areas covered by “gracile fasciculus and cuneate fasciculus” be labelled “Dorsal Column”; and “Ventral Horn” be used instead of “Ventral Column”.

REVIEWER COMMENTS

Reviewer #1 (Remarks to the Author):

The authors have addressed my points.
Thank you for the nice report.

Response: We thank you for your kind comments.

Reviewer #2 (Remarks to the Author):

The authors have dealt with my question about how the amount of peroxide generated by DAO used to cause the ataxia relates to the molecular species and their abundance in the the real world disease situation by adding some helpful Discussion text that reasonably addresses this issue. I think there is not much that can be done to fully tackle this question experimentally, and so I find the added text sufficient in the circumstances.

Response: We thank you for your thoughtful comments.

Reviewer #3 (Remarks to the Author):

I am impressed with the thoroughness with which the authors responded to comments from all reviewers. As I noted previously, I considered this work is of the highest caliber and eminently suitable for publication in Nat Comm. They have addressed my concerns, adding relevant data and altering their discussion to focus on the specific outcomes of their experiments.

Response: We thank you for your kind comments.

Reviewer #4 (Remarks to the Author):

The authors have responded satisfactorily to the comments of this reviewer. To rephrase the notion that the transgenic/chemogenetic approach could lead to "an informative new model of Friedreich's ataxia" to "new insights into the molecular pathogenesis of Friedreich's ataxia" has resolved the critical concerns on this approach.

Response: We thank you for your thoughtful comments.

Two minor suggestions:

1. Lines 108-109 and 401: To conform with Panels c and d in Extended Figure 1, it is

recommended that “dorsal column” and “ventral horn” be used instead of “dorsal tracts” and “ventral (motor) tracts”.

Response: We have changed the text in Lines 108-109 and line 401 as suggested.

2. Figure 1a: Note that “dorsal horn” is NOT part of the “Dorsal Column” and should not be included in the "box"; the term “Ventral Column” is anatomically ambiguous. Again, to conform with Panels c and d in Extended Figure 1, it is recommended that only areas covered by “gracile fasciculus and cuneate fasciculus” be labelled “Dorsal Column”; and “Ventral Horn” be used instead of “Ventral Column”.

Response: We have modified Figure 1 as suggested.

REVIEWERS' COMMENTS

Reviewer #4 (Remarks to the Author):

1. I regret to note that by simply removing "Column" from "Dorsal Column" and "Ventral Column" in Figure 1a, the authors "have NOT modified Figure 1 as suggested". For Figure 1b,c to be easily comprehended by the readers, it is strongly recommended that "Dorsal Column" must be clearly denoted in Figure 1a; "Ventral" should be deleted.

2. Lines 103-104: "the dorsal column....had degenerated. In contrast, the ventral column was apparently unaffected (Figure 1a-1c)" should read "the dorsal column....had degenerated (Figure 1a-1c). In contrast, the ventral horn was apparently unaffected (Extended Figure 1c,d)".

a. As previously suggested, "ventral column" is anatomically ambiguous.

b. "Figure 1a-1c" do not contain data on ventral horn,

c. "Extended Figure 1c,d" is never mentioned in the main text.

3. Extended Figure 1c,d: Label "Ventral Horn".

REVIEWER COMMENTS

Reviewer #4 (Remarks to the Author):

“1. I regret to note that by simply removing “Column” from “Dorsal Column” and “Ventral Column” in Figure 1a, the authors “have NOT modified Figure 1 as suggested”. For Figure 1b,c to be easily comprehended by the readers, it is strongly recommended that “Dorsal Column” must be clearly denoted in Figure 1a; “Ventral” should be deleted.”

We have done as suggested by the Reviewer.

“2. Lines 103-104: “the dorsal column....had degenerated. In contrast, the ventral column was apparently unaffected (Figure 1a-1c)” should read “the dorsal column....had degenerated (Figure 1a-1c). In contrast, the ventral horn was apparently unaffected (Extended Figure 1c,d)”. “

We have edited as suggested by the Reviewer.

“a. As previously suggested, "ventral column" is anatomically ambiguous. “

We have changed “ventral column” to “ventral” to show orientation of the section

“b. “Figure 1a-1c” do not contain data on ventral horn.”

We have corrected this as suggested.

“c. “Extended Figure 1c,d” is never mentioned in the main text.”

We now mention this Extended Figure in the text (line 105).

“3. Extended Figure 1c,d: Label “Ventral Horn”.”

We have changed the label to “Ventral” to indicate the orientation of this image.